

# Unravelling the wind impact of clusters of storms, a case study over the French insurer Generali

Laura Hasbini[1,2], Pascal Yiou[1], Laurent Boissier[2,3], and Arthur Perringaux[2]

[1]Laboratoire des Sciences du Climat et de l'Environnement, UMR8212 CEA-CNRS-UVSQ, U Paris-Saclay & IPSL, 91191, Gif sur Yvette, FRANCE
[2]Generali France, 93210, Saint Denis, FRANCE
[3]Univ Paul Valéry Montpellier, LAGAM, F34000, Montpellier, FRANCE

**Correspondence:** Laura Hasbini (laura.hasbini@lsce.ipsl.fr)

**Abstract.** Winter windstorms cause extensive damage to infrastructure and represent the most significant natural hazard for Generali France in terms of insured losses. This study presents a method to systematically link physical storm events with observed insurance claims, enabling a better understanding of which storms, including weaker depressions, drive losses within Generali's portfolio. The proposed association represents a cornerstone for the calibration of insurance and reinsurance processes such as risk assessment, loss modelling and prevention. Beyond analysing individual events, we assess the impact of storm clusters, defined as multiple storms affecting the same region within a 96-hour window, consistent with reinsurance contract definitions. Our findings reveal that 85% of windstorm-related losses since 1998 are attributable to clustered events. The most intense storms are frequently preceded or succeeded by smaller, yet damaging, depressions. This is illustrated by the case of Storms Anatol, Lothar and Martin in December 1999 and Storm Klaus in January 2009. Furthermore, we find that storms causing damage are more likely to occur as part of a cluster (50%) compared to the overall population of depressions affecting France (29%). These findings highlight the importance of accounting for storm clustering in risk modelling and reinsurance strategies.

## 1 Introduction

Extratropical cyclones (ETC) are a dominant meteorological phenomenon in the mid-latitudes, serving as key drivers of day-to-day weather and accounting for the majority of high-wind and precipitation events across Europe (Hawcroft et al., 2012). Windstorms are intense ETC associated with extreme wind events. In Europe, they rank among the most costly natural hazards and are estimated to cause economic losses averaging 193€ billion per winter (ECMWF, 2024). Given their significant societal and economic impacts, windstorms have gathered considerable attention from climate sciences, meteorology, and insurance sectors. Physically, they are characterised by storm tracks that follow the displacement of the ETC. While various tracking algorithms effectively capture the general patterns of storm tracks, the choice of methodology significantly influences the number and characteristics of detected ETCs (Neu et al., 2013). For well-developed storms, tracking algorithms tend to produce very consistent results. However, discrepancies arise for weaker, slower-moving, or short-lived ETCs (Flaounas et al., 2023). To quantify the impacts of windstorms on society, several metrics have been developed. The most common one is the




Storm Severity Index (SSI) introduced by Klawa and Ulbrich (2003). The SSI has been extensively applied in recent studies
(Leckebusch et al., 2008; Lockwood et al., 2022; Priestley et al., 2023; Little et al., 2023; Cornér et al., 2024), providing a
robust framework for assessing storm intensity and associated risks. By analysing the full lifecycle of ETC, key physical char-
acteristics linked to more intense storms can be identified. These include explosive cyclogenesis (Ludwig et al., 2015; Ginesta
et al., 2023) and the presence of a strong jet stream (Hillier et al., 2025). While such features are critical for understanding the
broader dynamics of ETC, they alone are insufficient for accurately predicting local-scale impacts.

In Europe, ETC frequency does not follow a Poisson distribution with a constant intensity (Mailier et al., 2006). This means
that the number of ETC observed at a given location and for a given period (ranging from a season to a few days) can vary
greatly. The possible temporal concentration of cyclones at a given location is called cyclone serial clustering (Dacre and
Pinto, 2020). Variations in cyclone counts can, for example, be attributed to large-scale atmospheric dynamics or interaction
between cyclones. Persistent conditions, such as an intense and zonal jet stream, promote Rossby Wave Breaking (RWB),
which can lead to the serial clustering of cyclones (Pinto et al., 2014; Priestley et al., 2017b). Regions such as Western Europe,
located at the exit and flanks of the North Atlantic storm track, are particularly susceptible to cyclone clustering (Dacre and
Pinto, 2020). The concept of "cyclone families," first introduced by Bjerknes and Solberg (1922), further explains how the
trailing conditions of a primary cyclone can facilitate secondary cyclogenesis through moist processes. Recent studies (e.g.
Pinto et al., 2014; Priestley et al., 2020) have demonstrated that secondary cyclones dominate during clustered periods over
Western Europe, underscoring the importance of these dynamics.

Serial clustering can be described by various metrics, with aggregation periods ranging from days to seasons. One commonly
used is the dispersion statistic, defined as $\psi = \frac{\sigma^2}{\mu} - 1$, where $\mu$ and $\sigma$ represent the mean and variance of cyclone counts over
a given time interval (Mailier et al., 2006). While this relative frequency metric has the advantage of not being dependent on
the local storm frequency, it does not provide globally comparable information suitable for impact assessment. In contrast,
absolute frequency metrics, such as the count of cyclones over a fixed region and period, offer globally comparable data but
are sensitive to the specific set of storm tracks used (Pinto et al., 2014, 2016). The choice of metrics depends on the research
objectives. From an event and impact perspective, an absolute definition with a 96-hour window is particularly relevant for
capturing the temporal clustering of extreme events.

Clustering is especially pronounced for extreme cyclones in Western Europe (Vitolo et al., 2009). Notable examples include
the winters of 1989/1990 (storms Daria, Herta, Nana, Judith, Ottilie, Hilie, Polly, Vivian, and Wiebke), 1999/2000 (storms Ana-
tol, Lothar, and Martin) (Rivière et al., 2010), and 2013/2014 (storms Christian, Xavier, Dirk, Anne, and Christina) (Priestley
et al., 2017a). These events, which caused significant damage, highlight the amplified risk associated with cyclone clustering
and underscore the need for improved understanding and prediction of such phenomena.

The insurance industry relies heavily on the representation of hazard, as this underpins risk assessment, loss modelling and
prevention. Enhancing the accuracy of hazard representation directly improves the overall efficiency and reliability of the in-
surance and reinsurance processes. Estimating storm impacts requires understanding the interplay between hazard, exposure,
and vulnerability (Intergovernmental Panel On Climate Change, 2023). From this perspective, storms are often defined using
surface wind intensity aggregated over multiple days (Moemken et al., 2024a; Severino et al., 2024). When combined with



insurance loss data, this approach develops more robust damage models and vulnerability curves, which are essential tools for
estimating expected losses (Prahl et al., 2015; Fonseca Cerda et al., 2024). However, this method presents key limitations: it
does not capture the detailed development of storms or the specific characteristics driving damage. Additionally, the temporal
aggregation of wind data obscures the contributions of individual storms within clustered events, making it difficult to distin-
guish their respective impacts. As a result, current models often struggle to adapt to clustered events, where overlapping or
sequential storms generate complex loss patterns that are not easily captured by traditional approaches.

Several datasets link storm damage to meteorological events, including the Extreme Wind Storms (XWS) Catalogue (Roberts
et al., 2014), PERILS (PERILS, 2025), the Copernicus Climate Change Service (Copernicus Climate Change Service, 2020)
and Munich RE (Munich RE, 2025). While these datasets provide valuable insights, they often differ in the identified storms
as well as in their loss estimates (Moemken et al., 2024b; Flynn et al., 2024). Designed primarily for reinsurance purposes,
they focus on the most impactful storms and provide smoothed loss estimates at regional or national scales. This limits their
usefulness for detailed analyses linking storm intensity to localised damage. Working with primary insurance data, such as
Generali France, allows for a more granular understanding of damage patterns, capturing a broader range of storm-related
claims. Nonetheless, this raises the fundamental challenge of defining storm events based on damage databases (Kron et al.,
2012). Stucki et al. (2014) demonstrate that temporally aggregated impacts can be extracted from such databases, though they
are subject to biases, particularly those stemming from historical reporting practices and exposure variations. Despite these
challenges, leveraging high-resolution claims data enables a precise damage assessment, distinguishing not only major storms
but also individual members within storm clusters, offering a more refined perspective on impact patterns.

This study presents a method for linking high-resolution insurance claims to storm events, unlocking several advantages.
From a vulnerability perspective, it can provide detailed insights into the structural and environmental factors contributing to
damage. From a meteorological standpoint, it disentangles the impacts of successive storm events, significantly enhancing our
understanding of storm clustering effects. Accurately associating claims with specific storms is essential for impact research as
it forms the basis for estimating storm-related costs, identifying return levels, and calibrating vulnerability curves for different
exposure scenarios. This paper addresses the following key questions :

- How can high-resolution insurance claims data be reliably linked to ETC?

- How can impact data be leveraged to distinguish the impact of successive storms?

- What is the impact of short-duration storm clusters on insurance losses incurred by Generali France?

The paper is structured as follows. Section 2 characterises the dataset of Generali's claims, ETC tracks, and the method to
identify clusters. The association method is described in Section 3 with a sensitivity analysis and evaluated by comparison
with other datasets. Section 4 examines the results of the association in the context of serial clustering, with two case studies
over storms Lothar and Martin (Dec. 1999) and storm Klaus (Jan. 2009). Section 5 ends the paper with some perspectives and
conclusions in Section 6.



## 2  Data

### 2.1  Storm data

The study is performed using ERA5 historical reanalysis (Hersbach et al., 2020). The dataset has a horizontal resolution of $0.25° \times 0.25°$, corresponding to 18 km at $50°$ N, and covers the period $1979 - 2024$ from October to March. Over such a dataset, storm tracks in the Northern Hemisphere were identified using the object-oriented algorithm TRACK algorithm (Hodges, 1999). First, the input data is smoothed over a $T42$ resolution (approximately $2.8°$) to filter out potential noise. ETC are then identified based on $6-$ hourly values of relative vorticity at 850 hPa connected with nearest neighbour search. No conditions are set on the displacement of trajectories nor the minimal value of the maximal vorticity. We apply a spatial constraint to retain only storms impacting France. This is done by selecting the tracks with a maximal distance of 1300 km from the territory of metropolitan France. The radius of 1300 km ensures that the largest depressions, which can have a distant impact, are not discarded. Additionally, only the well-developed depressions lasting more than 24 h are kept. This leads to a set of 4439 storms over the period ranging from March 1979 (included) until March 2024 (excluded).

For each identified track, we define the storm landing date ($d_{\mathrm{storm}}$) as the date when the track is the closest to the longitude line of $7.5°$ W. This point of closest approach is used as a reference to assign a unique identifier to each storm. The storm is named using the landing date and time, along with the latitude and longitude of its closest point. For example, Storm Lothar reached its closest point to the $7.5°$ W line on 26 December 1999 at $12 : 00$ UTC, located at $4.2°$ W and $51.5°$ N. It is therefore labeled as: *"1999-12-26 12h [-4.2;51.5]"*. This naming convention has been shown to uniquely identify storms, even in cases where multiple storms made landfall on the same date, by incorporating both temporal and spatial characteristics of their trajectories. Storm footprints are defined using hourly $10\mathrm{m}$ wind gust with a circular spatial mask of $1300\mathrm{km}$ radius around the centre of maximal vorticity and a temporal window of $\{-12, \ldots, 12\}$ hours around each tracked point. Additionally, we defined the impact area of a given storm with a radius of $r$ km moving with the centre of maximal vorticity.

The TRACK scheme has been shown to effectively represent ETC and is well-suited for impact analysis. However, compared to other studies (Priestley et al., 2024; Lockwood et al., 2022), we relaxed the conditions on cyclone duration and intensity. This was done to ensure the inclusion of more ETCs, particularly fast-moving and smaller-scale storms, which can be responsible for significant damage locally. Priestley et al. (2020) also showed the importance of secondary cyclogenesis in serial clustering, which are often small-scale storms. Capturing storms of all scales is thus key for understanding damage during clustered events. The analysis is performed using a single tracking, which can present some limitations (Neu et al., 2013). The capture of depressions of all scales reduces the bias incorporated by the choice of the tracking scheme.

### 2.2  Serial clustering of storms

Definitions of storm clusters vary across the literature and typically depend on both temporal and spatial criteria. Previous studies have employed a range of approaches, including absolute frequency metrics (Pinto et al., 2014; Karwat et al., 2023; Hauser et al., 2023), relative frequency metrics (Mailier et al., 2006; Economou et al., 2015; Pinto et al., 2016), and different temporal windows (Dacre and Pinto, 2020). In this study, we adopt an absolute frequency-based approach, complemented by a




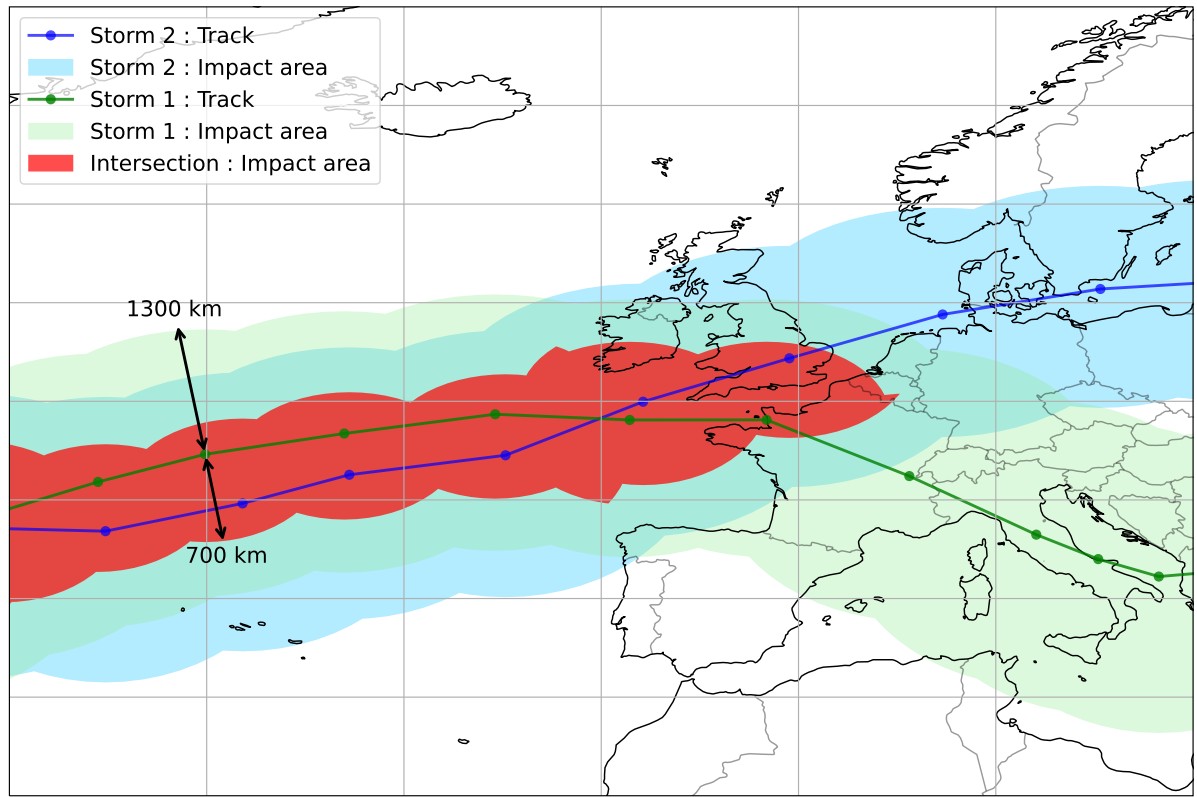

**Figure 1.** Example of clustering with two storms. The green and blue lines represent the storm tracks of storms 1 and 2, with 6-hourly time increments. The green and blue shadings represent, respectively, the impact area of storms 1 and 2, defined with a radius of 1300 km around the centre of the track. The red shading illustrates the intersection of "high-impact" areas, different by a radius of 700 km around the storm tracks.

spatial criterion. Specifically, two or more storms are considered part of the same cluster if their 700km-impact areas intersect, and their landing dates are separated by no more than 96h. The choice of a 700 km radius is based on the 6°angular distance around the centre of maximum vorticity, which has previously been used to define the region of strongest wind impacts (Zappa et al., 2013; Gramcianinov et al., 2020; Cornér et al., 2024). This angular distance corresponds to approximately 70 km at latitudes up to 55° N. The 96-h temporal window is consistent with storm definitions used in Generali's reinsurance contracts. To focus on events relevant to the French territory, we further require that the impact area intersection occurs over France. In cases where one identified cluster is entirely contained within another, the smaller (subset) cluster is discarded.

Under this definition, a single storm can be part of several clusters. Among the 4439 storms affecting France, 1283 are part of at least one cluster. While one storm is found to participate in as many as 4 distinct clusters, the average is 1.18 clusters per storm. In total, 517 storm clusters are identified, with an average of 2.7 storms per cluster and a maximum of 13 storms in a single cluster.



## 2.3 Insurance data

Claim data from Generali's portfolio, spanning from 1998 to 2024, is used as the primary impact metric. From the most recent analysis of building insurance in France, the market share of Generali in France can be estimated at 3% (Frédération France Assureurs, 2024a, b). Generali's claims data provides a robust representation of storm-related insurance impact. The analysis focuses on windstorm claims recorded by the Generali IARD entity (Fire, accidents and miscellaneous risks) during the extended winter season (September-April). Each claim is characterised by the latitude and longitude of the impacted building, the date of declaration and the date of damage. In this study, we refer to the estimated date of damage as the claim date ($d_{\text{claim}}$). Damage intensity is quantified by the net insured loss, over which deductibles are truncated. As a result, some values may be zero or negative; these entries are excluded from the analysis. In addition, claims exceeding 150000€ in net loss at the date of damage, which correspond to severe damage, are treated through a separate process and are also excluded. To ensure consistency across years, all monetary losses are detrended and converted to constant 2015 euros, using inflation indices provided by the Institut National de la Statistique et des Etudes Economiques (INSEE, 2025). Claims can be categorised as closed, open or out of order. Closed claims have been reimbursed based on the reported loss. Open claims are still under assessment, with losses subject to revision. Claims labelled as out of order, meaning not attributable to the relevant hazard, are excluded from the dataset. The full dataset was extracted on the $1st$ October 2024 and contains approximately 3% of opened claims, primarily corresponding to events in March and April 2024, at the end of the 2023/2024 winter season. After applying the filtering mentioned above, the cleaned dataset contains 210435 entries.

The geographical locations of insured properties are derived from textual addresses and converted into latitude and longitude using the BD TOPO (Institut national de l'information géographique et forestière, 2025), a vector database that compiles all French addresses. The geocoding tool from the Environmental Systems Research Institute (ESRI, 2025) is used to match addresses to this reference. Among the 210435 reported claims, 12.7% have an "excellent" location match, meaning both the exact building and its characterisation are identified. 55.1% are linked to the correct building but without the building's features, categorised as "good" match. For 22.6% of cases, only the street or hamlet name is identified, classified as "average" geocoding. Lastly, $\approx 10\%$ have a "bad" geocoding quality, with locations only determined at the postcode level.

## 3 Methods

### 3.1 Association of claims to storm events

Claim dates filled in the damage dataset are inherently biased as they are reported by policyholders based on their perception and interpretation of the hazard. Known as historical perception, the raw loss dataset tends to over-represent intense windstorms and under-represent the ones of smaller intensity (Stucki et al., 2014). This is exacerbated by the identification of storm dates, which cannot be used as a reliable identifier without preprocessing. The usual strategy is to aggregate the damage over several days, as done by Stucki et al. (2014), but this erases the contribution of smaller-scale storms. To address this data quality issue



while still capturing the impact of all ETC events, we propose an association strategy that maps the claims and the storms accurately.

The first step consists of linking each claim to all storms with a landing date close to the claim date. This closeness is determined by a temporal window defined by several days before ($X_b$) and after ($X_a$) the storm date ($d_{\text{storm}}$). Concretely,

storms are selected so that their landing date $d_{\text{storm}}$ verifies $d_{\text{storm}} - X_b \leq d_{\text{claim}} \leq d_{\text{storm}} + X_a$ (in days). Here, $d_{\text{claim}}$ is the date of occurrence of the claim. This initial selection may link a claim to one or multiple storms. Since the ultimate goal is to associate each claim with a single storm, the storm most likely resulting in the damage will be chosen following a method based on wind gust value. Each claim is linked to the storm with the highest wind gust value at the claim location. Additionally, following the understanding of storm damage and Generali's exposure, it is unlikely that a storm will result in only few claims.

To overcome this, we set a minimal number of claims ($n_{\text{claims}}$) that a storm should be linked to. If a storm is associated with fewer claims than this minimum, its damage is reassigned to the closest storm in terms of date. This process is repeated iteratively, increasing the number of claims by 10 each time, until robust and reliable results are achieved.

The proposed association depends on three tuning parameters $X_b$, $X_a$ and $n_{\text{claims}}$, respectively corresponding to the number of days before the storm date, the number of days after the storm date and the minimal number of claims associated with a

storm. These parameters help fine-tune the association between claims and storms. The performances of the association are evaluated with three metrics, comparing the identified storms to the claim's local maxima. The local maxima are identified by peaks over the time series of claim count gathering at least 10 claims. The tuning of the association assumes that the loss data correctly captures the major physical events, although they can present some temporal shift. The local maxima identified using the distribution of the number of claims as a function of time represent the target of the association. The constructed metrics

evaluate whether the number of storms corresponds to the number of peak claims and whether the claim peak dates are not too far from the actual storm landing date. The precision metric ($M_{\text{days}}$) is computed as the maximal difference between a storm date and its closest local maximum. The smaller this difference, the better the association. The frequency metric ($\Delta_f$) is defined as the difference between the number of storms compared to the number of local maxima. Lastly, a completeness metric ($P_{claims}$) corresponding to the percentage of claims associated with storm events is defined. These metrics collectively provide

a robust evaluation of how well the association strategy maps claims to storms, ensuring both accuracy and completeness in the analysis. No spatial performances are evaluated at this stage. It is assumed that the conditioning on the storm with the highest wind gust should already capture the spatial distribution.

In mathematical terms, let $\{S_1, \ldots, S_n\}$ be the dates of the storms identified and $\{L_1, \ldots, L_m\}$ the dates local maxima. Then the frequency difference is defined as $\Delta_f = n - m$. The precision metric can be written as $M_{\text{days}} = \max\limits_{j \in [1,n]} \min\limits_{i \in [1,m]} |S_j - L_i|$. The

difference between the dates $|S_j - L_i|$ is expressed in days as the local maxima computed over the claims dates are at the day level.

## 3.2    Sensitivity of the association to parameters

This subsection inspects the evolution of the performance metrics as a function of the tuning parameters. The performance metrics are computed independently for each winter and for $X_b$ in $\{0, 1, 2, 3\}$, $X_a$ in $\{2, 3, 4, 5\}$ and $n_{\text{claims}}$ between 30 and





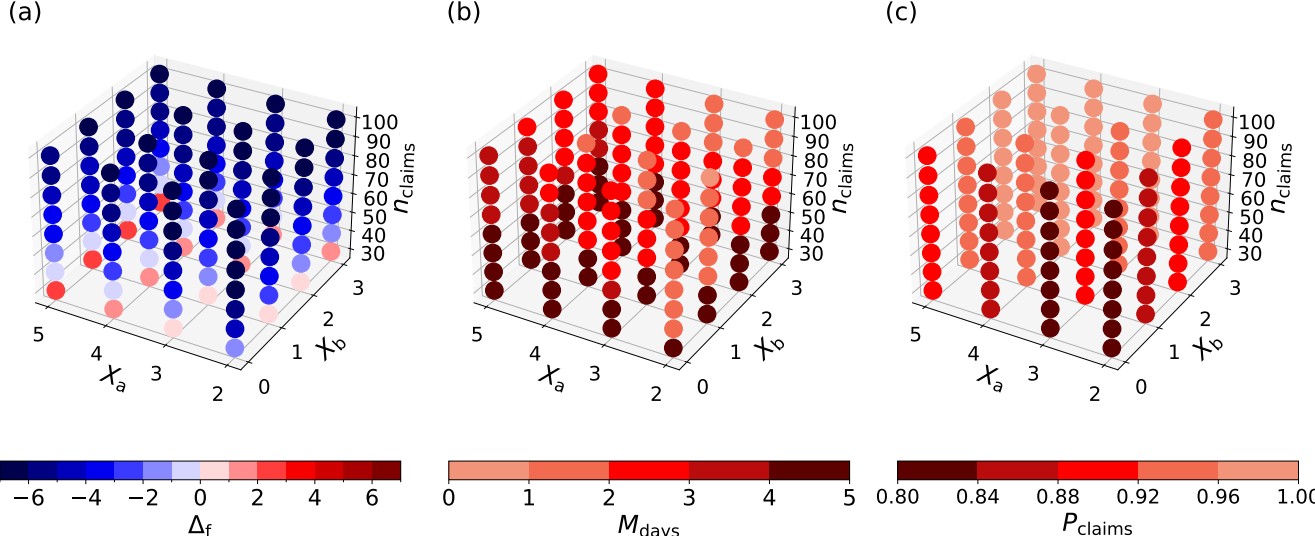

**Figure 2.** Frequency ($\Delta_f$) in number of events (a), precision ($M_{\mathrm{days}}$) in number of days (b) and completeness ($P_{claims}$) in percent (c) as a function of $X_a$ (horizontal $x$), $X_b$ (horizontal $y$) and $n_{\mathrm{claims}}$ (vertical $z$). The optimal set of parameters($X_a$, $X_b$, $n_{\mathrm{claims}}$) is obtained when the difference is null.

100 with 10 claims steps. The metrics are then averaged over all the winters. Figure 2 shows the evolution of the frequency, precision and completeness metrics as a function of the tuning parameters. Lighter colours indicate better results.

The impact of the association windows can first be analysed. If $X_a$ is too large, the precision decreases with a bigger minimal difference between the storm date and the closest peal date (Fig. 2b). Conversely, increasing $X_a$ enables to capture of more claims, thus being more representative of the global dataset (Fig. 2c). In terms of performance, this improves the completeness metric. Nonetheless, we see that the variation in the completeness varies less than the precision metric. The behaviour of $X_b$ is similar to that of $X_a$. A greater value would allow for more completeness, but would also decrease the precision of the association. The $n_{\mathrm{claims}}$ parameters do not alter the completeness performances. This parameter defines the threshold under which claims should be shifted; it thus does not influence the number of claims linked to some storms. Regarding precision (Fig.2b), shifting claims reduces the maximal difference between storm dates and the nearest peaks. This underlines that the shifted claims seems to be those farthest from the peak dates

The behaviour of the frequency metric is more complex as it depends on the interaction of the 3 parameters (Fig. 2a). For a given window, increasing the minimal number of claims ($n_{\mathrm{claims}}$) reduces the number of storms and consequently decreases the $\Delta_f$. This reduction is needed when a positive frequency is observed. When more storms are detected than the number of local maxima. However, decreasing it too much leads to a negative one. This corresponds to over-concentration around the major storm events, with more local maximums than the number of storms. The situation when too many storms are captured (positive $\Delta_f$) is mostly observed when $n_{\mathrm{claims}}$ is small.



A trade-off should be found between the size of the association windows ($X_a$ and $X_b$) and the strength of the concentration over the major events ($n_{\text{claims}}$). The optimal association should not only be complete but also associate the claims with their correct storms. Fig. 2c shows that the smallest percentage of claims that can be linked to storm events is $80\%$. As this is already satisfactory, the quality of the linkage will only be measured using the difference in frequency and the maximal temporal difference between storm events and local maximums.

The optimal tuning parameters $X_b$, $X_a$ and $n_{\text{claims}}$ are found by minimizing the cost function defined as :

$$f_{cost}(w_{freq}) = \sqrt{w_{freq} \times \Delta_f^2 + (1 - w_{freq}) \times M_{\text{days}}^2} \tag{1}$$

The frequency and precision metrics are centred between 0 and 1, $w_{freq}$ is a weight attributed to the frequency metric, which can vary between 0 and 1. This weight is used to quantify the sensitivity of the optimal parameters found. Optimisation is performed over $f_{cost}$ using global minimization search for $(X_a, X_b, nb_{\text{min\_claims}}) \in \{0,3\} \times \{2,5\} \times \{30,100\}$. The robustness of the results to $w_{freq}$ is developed in Appendix A.

Minimising the loss function leads to $X_b = 3$, $X_a = 3$ and $min\_claims = 50$. Such parameters will be kept for the rest of the study. The obtained window of 7 days is comparable to the one of 5 days used by Fonseca Cerda et al. (2024). We expect that such large windows will better account for the potential postponed impact of a windstorm. Additionally, the linking strategy based on the maximal wind-gust value should ensure the association with the correct storm driver.

### 3.3 Representativity of the catalogue

Assessing the representativeness of the resulting catalogues remains a methodological challenge due to the variety of available approaches. In a comparison between academic and insurance-based catalogues, Moemken et al. (2024b) highlight significant discrepancies in storm frequencies, major event identification, and estimated losses. These differences stem from the primary objectives of each catalogue but may have important implications for risk assessment. The proposed approach, especially tailored to Generali's exposure, should result in the most precise and accurate catalogue.

A key consideration is the dataset ability to accurately capture the most severe historical events. Applying the association method described above, 344 storms are associated with an impact for Generali over the $1997-2024$ period. Among these, the 40 costliest events are presented in Fig. 3. Notably, storms Anatol, Lothar, Martin, Klaus, Xynthia, Eunice, and Ciaran were the most costly for Generali, with losses exceeding $10M€$ per event. Importantly, our method successfully captures high-impact storms documented in global datasets (PERILS, 2025; Flynn et al., 2024; Copernicus Climate Change Service, 2020), as evidenced in Fig. 3. Further comparison with Météo France's catalogue, produced using the SSI, reveals strong agreement over the period $1998-2024$ (Meteo France, 2023). Only the storms of 16-17 December 2019, storm Andrea and storm Calvann, rank among the most intense in Météo France's analysis, but do not appear in the top 40 costliest events from Generali. This discrepancy warrants further investigation, with one plausible explanation being the influence of Generali's portfolio.

The resulting catalogue exhibits a great yearly variability in both the number of impacting storms and in losses amount. Fig. 4 underlines that the costliest winter seasons do not always correspond to the year with the most numerous storms or clusters. This is especially the case for the winter $1999/2000$ with only 7 storms associated with damage, but corresponding to the





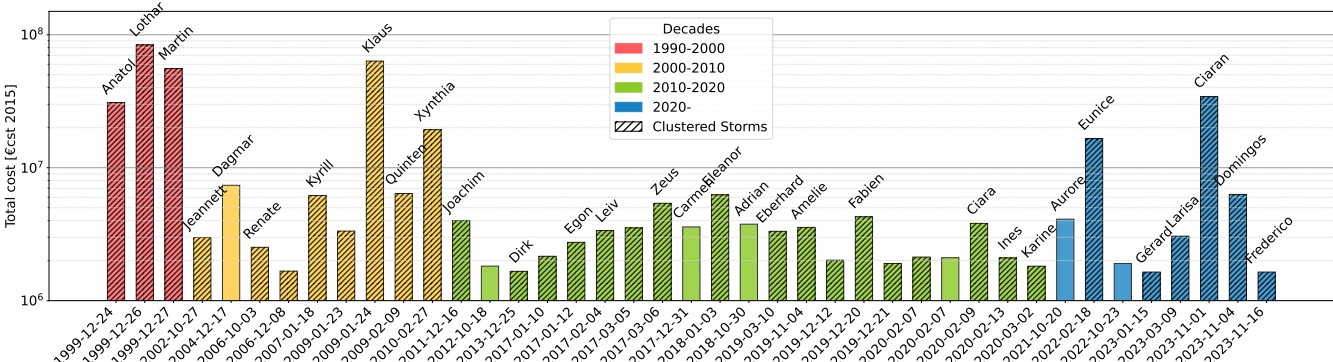

**Figure 3.** Losses in constant € 2015 of the 40 costliest storm for Generali over the period $1998-2024$. Storm dates are indicated in the $x$-axis and Storm names over the bar plot of each event. Colours indicate the decade during which the storm event occurred. Dashed bar indicates storms being part of a cluster.

costliest winter. The winter $2006/2007$ is also notable with 23 storms identified with damage, but a total loss for the season close to the average. This high number of storms associated with moderate losses can come from the diversity of our storm tracks. As small and fast-moving depressions were kept in the set of storm tracks, it is more likely to encounter small storms associated with little losses.

      The pronounced interannual variability in the number of ETCs is consistent with the intrinsic characteristics of these systems,
as highlighted by Feser et al. (2015). Furthermore, Moemken et al. (2024b) emphasises that this variability can become even more pronounced when considering storm-related losses, due to their strong dependence on varying exposure levels from year to year. The patterns of interannual variability in both storm counts and associated losses, as illustrated in Fig. 4, are therefore in line with previous findings in the literature. We can underline that the variability of losses varies with both the number of storms, their intensity, but also the local vulnerability. This underscores the complex interplay between meteorological
variability and socioeconomic vulnerability.

      The quality of the dataset can be evaluated using insurance-based metrics. The catalogue should reproduce the global damage statistics of storms. Fig. 5 illustrates the distribution of the total cost per storm, the mean cost per claim and the number of claims per storm. Such metrics are computed after aggregating policy-based costs at the storm level. An important shift can be found between the mean and median for both the total cost (Fig.5a) and the number of claims per storm (Fig.5c). Such shifts are
related to the heterogeneity of storm costs. The most intense storms, associated with long return periods, correspond to outliers of the main distribution. Frédération France Assureur (2025) estimated a mean cost of varying between 1530 and 2335 in € cst of 2023 for residential properties. This corresponds to a range of $[1285, 1962]$ in constant € of 2015. These values are lower than our mean estimate of 2790€ in Fig. 5. Mission Risques Naturels (2021) performed an in-depth study over the restricted winter $2018/2019$. Their results underline that the average cost greatly depends on the type of infrastructure impacted. The
gap with the observed mean damage in Fig. 5 can also be explained by the exposure of Generali as well as the historical depth. Generali is mostly implanted in the cities (Paris, Lyon, Lille, Bordeaux), the South-East of France and in the North.





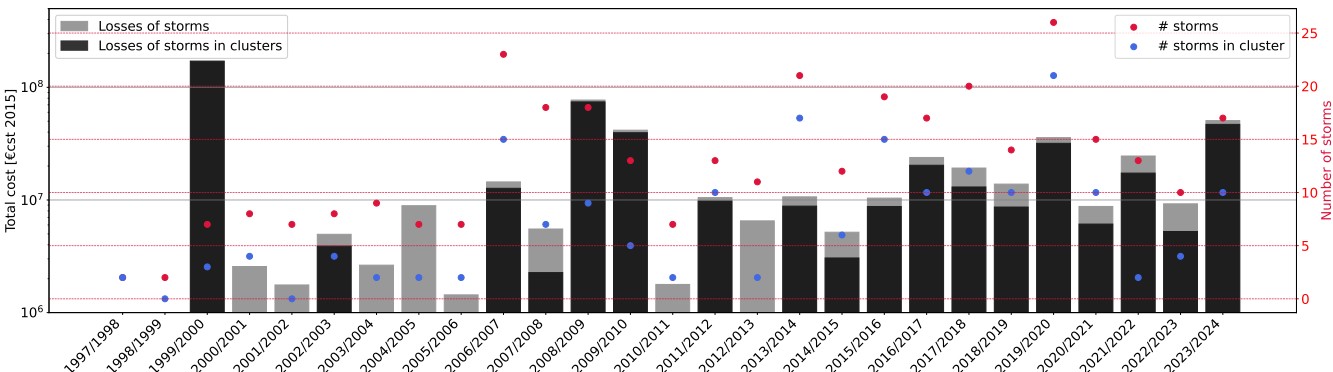

**Figure 4.** Yearly insured losses and storm occurrences. Light gray bars represent the total losses in constant € 2015 per winter season; darker bars indicate the amount linked to clusters of storms. The points represent the number of storms (red) and the number of storms being part of clusters (blue) per winter season. Winter $y - y + 1$ start from September of year $y$ and ends in March of year $y + 1$

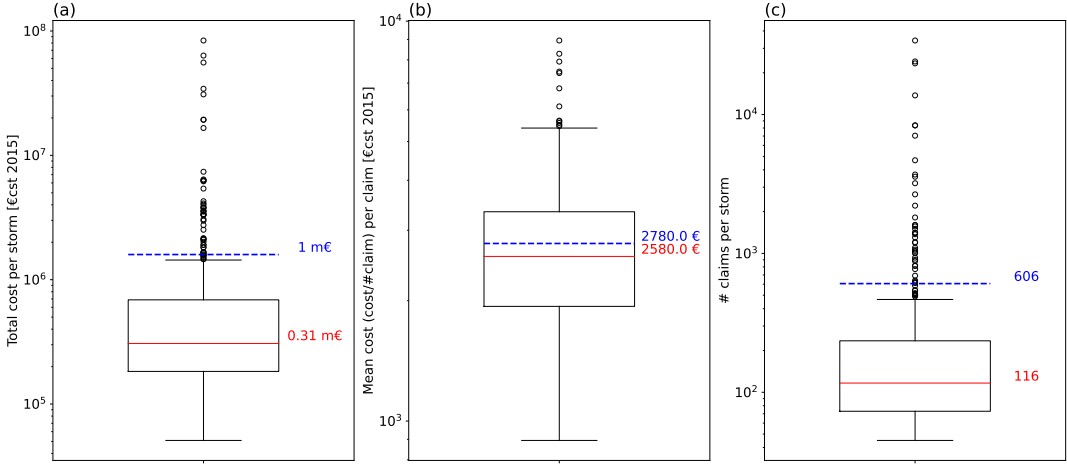

**Figure 5.** Distribution of the total cost per storm in constant € 2015 (a), the cost per claim in constant € 2015 (b) and the number of claims per storm (c). Red lines indicate the median, and blue dashed ones the mean.





## 4   Results : Impact of storm clustering

### 4.1   Global statistics of storm clustering impacts

The clustering method described in section 2.2 can be applied to the set of impacting storms. This identifies events such as
storms Lothar and Martin in 1999, when at least two storms were responsible for damage. Over the ensemble of 344 impacting
storms, we found that 176 of them are divided into 82 high-impact clusters. This corresponds to a ratio of 50%, greater than
the one of 29%, found when applying clustering over all the storm tracks. It means that the most impactful storms are more
frequently part of clusters. Although such storms are not the most numerous, they result in most of the damage. It can already
be seen in Fig. 3 that from the 40 most costly storms for Generali, 82% are part of a cluster. Extending this to the full dataset,
we found that clusters of storms are responsible for 85% of the total windstorm loss and 87% of the declared claims, which is
significantly greater than their physical occurrence frequency. Storms in clusters thus seem to be more frequently impacting,
but also more costly.

Section 3.3 underlined that storm count and losses exhibit significant yearly variability. This is also the case for storm
clustering. The period from 2000 to 2005 experiences a few impactful storms for Generali, resulting in moderate losses.
Conversely, the last 10 years from 2013 to 2023 were associated with important storm losses and a particular importance of
clustering. Interestingly, the number of storms in clusters observed in each season is not the main driver of the loss. We can
note the winter 2021/2022 with only 2 storms in clusters, but accounting for most of the loss. As already pointed out in section
3.3, this is a signal that the damage associated with the smaller depression is also captured. During the winters of 1999, 2008,
2009, 2019 and 2023, which are the costliest in record for Generali France, almost all the losses are associated with storms in
clusters. Over each season major storms can identified such as storms Anatol, Lothar and Martin in 1999/2000, storm Klaus
in 2008/2009, storm Xynthia in 2009/2010, storms Amelie, Fabien, Ciara, Ines and Karine in 2019/2020 or storm Ciarán,
Domingos and Frederico in 2023/2024.

Figs. 3 and 4 together emphasise the importance of an event-based analysis. What drives the loss is not only the clustering
phenomenon but also the storm itself. To understand the importance of the clustering phenomenon in the repartition of the loss,
we investigate how the loss is distributed within a cluster. For each storm member of a cluster, we compute its share of the
loss as the total cost of one of the storm events divided by the total cost of the cluster. We can compare this value to the loss
and occurrence rank. A loss rank of 1 corresponds to the costliest member of the cluster; similarly, an occurrence rank of 1
corresponds to the earliest storm of the cluster.

From the 82 cluster events identified, Fig.6 shows the share of the loss as a function of these ranks. Fig.6a highlights a great
spread of the loss within the several clusters, the costliest storm can be responsible for between 41 and 99% of the total cluster
losses. On average, we found that the costliest storm of the cluster represents 72% of the loss of the total event. One storm
usually dominates the total cost. Nonetheless, this value, far from the 100%, underlines the need for insurance to group storms
around clusters. It means that the total observed damage can come from the temporal clustering of such events.

Fig.6b shows that the order of arrival of the storms is not a driver of the loss. The first landing storm is on average worth 40%
of the total cluster's loss, but this amount varies greatly from almost no impact to the total impact of the cluster. The same can





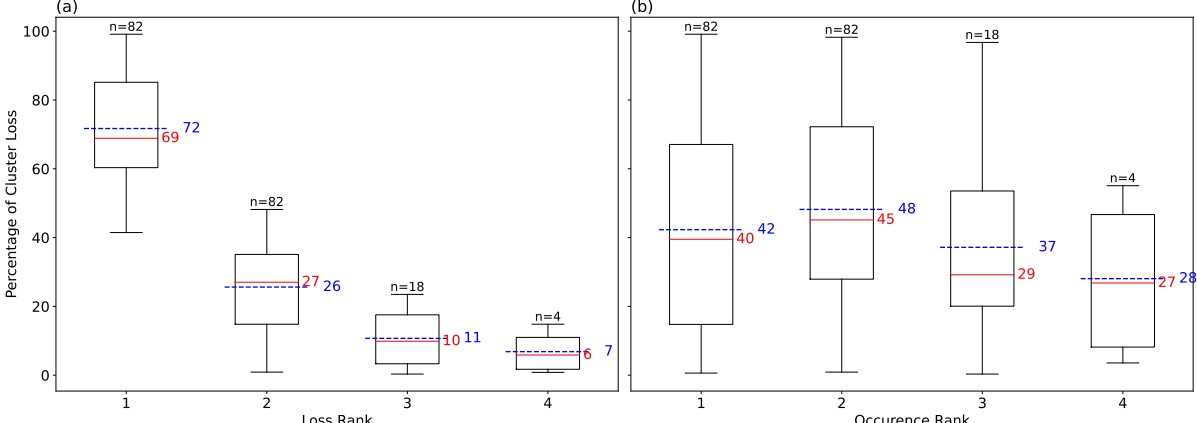

**Figure 6.** Distribution of the loss within the members of the cluster as a function of the loss (a) and occurrence rank (b). Thick red lines indicate the median, and blue dashed ones the mean. The numbers above each box correspond to the number of clusters used for each box plot.

be said for the $2nd$ landing storm. Lower values are found for the $3rd$ and $4th$ storms. However, it must be put in perspective that fewer cluster events are found with more than 2 storms. Additionally, an equally split loss corresponds to $30\%$ and $25\%$ of the total cluster loss when the cluster counts respectively 3 and 4 members.

We can note that the median and the mean are almost identical for all the box plots. This suggests the percentage of cluster loss follows a Gaussian distribution for all the given losses and occurrence ranks. The most extreme storms consequently do not exhibit specific statistics concerning these variables, as opposed to what was observed with event-based analysis in Fig. 5.

Among the 82 identified cluster events, Fig.6 shows how losses are distributed by storm rank. Fig.6 a reveals a wide variation in how much each storm contributes to total cluster losses: the costliest storm accounts for between $41\%$ and $99\%$ of total damages, with an average of $72\%$. This indicates that while a single storm often dominates the cost, it rarely accounts for the entire loss. This supports the relevance of grouping storms into clusters, as total losses often result from multiple temporally close events.

Fig.6b shows that the order in which storms occur within a cluster does not determine their financial impact. The first storm typically contributes $40\%$ of the total loss, but this varies widely, from negligible to the entire loss. Similar variability exists for the second storm. Third and fourth storms tend to contribute less, though fewer clusters contain more than two storms. For comparison, if losses were evenly split, the expected shares would be $30\%$ and $25\%$ for clusters with 3 and 4 storms, respectively.

Finally, the near-equality of the medians and means in the box plots suggests the distribution of cluster losses by loss and occurrence rank is approximately normal. Unlike the event-based results in Fig.5, the most extreme storms do not show distinct statistical behavior in this cluster-based view.



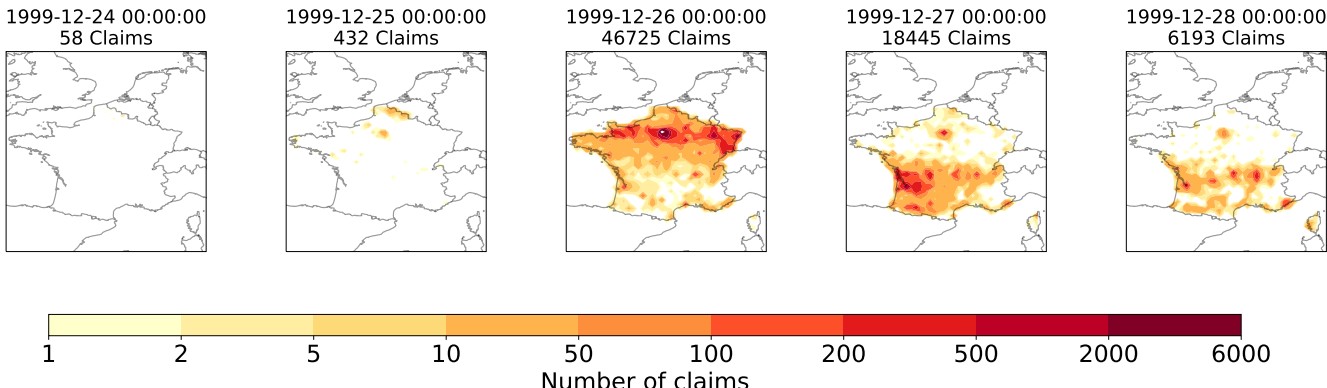

**Figure 7.** Map of the number of claims per day from $24/12/1999$ until the $28/12/1999$

A similar analysis can be applied to frequency (Fig.B1), represented by the number of claims per event. The results are comparable: on average, the most impactful storm accounts for $70\%$ of the total claims, with substantial variability in the share of claims across different claim ranks. Fig.B1 also indicates that the order in which storms occur within a cluster does not appear to influence how claims are distributed among the storms.

### 4.2   Case Study of December 1999: Storms Anatol, Lothar and Martin

The winter of $1999$ has represented a significant challenge for the insurance sector in Europe, particularly in France, due to the successive storms Anatol, Lothar and Martin on the $24th$, $26th$ and $27th$ December 1999. Storm Anatol had a northern trajectory, causing severe damage across Scandinavia (Kettle, 2021). Lothar, meanwhile, brought widespread destruction to France, Switzerland (Bründl and Rickli, 2002), Germany (Schmoeckel and Kottmeier, 2008; Schindler et al., 2009), and Belgium. Storm Martin followed a path similar to that of Lothar, leading to an exacerbated impact in the regions mentioned previously.

The presence of a strong upper-level zonal jet facilitated the formation of these storms (Wernli et al., 2002; Rivière et al., 2010). With only 24 hours separating them, their impacts are difficult to distinguish in aggregated data. At the time, the reinsurance 72 hour clause was loosened for recovery purposes and allowed for separating some losses (Risk Management Solutions, Inc, 2000). Nonetheless, this distinction was made solely for economic constraints and was not based on the actual damage caused by each storm. As a result, most studies treat Lothar and Martin as single events when evaluating their collective
impacts (Michèle Lai, 2019; Welker et al., 2021).

   Studies have underlined difficulties in disentangling the individual impacts of these storms. Fig.7, illustrating the spatial distribution of claims per day, highlights the impossibility of differentiating between storms Anatol, Lothar and Martin using claim date only. Fig.7 illustrates the spatial distribution of claims around the dates of both storm events. We can underline that claims were declared for each of the days and over the whole of France during the subset period. For cities such as Paris,
claims were identified each day. Without more information about the intensity and location of the storms, it is thus impossible to differentiate between the successive events.





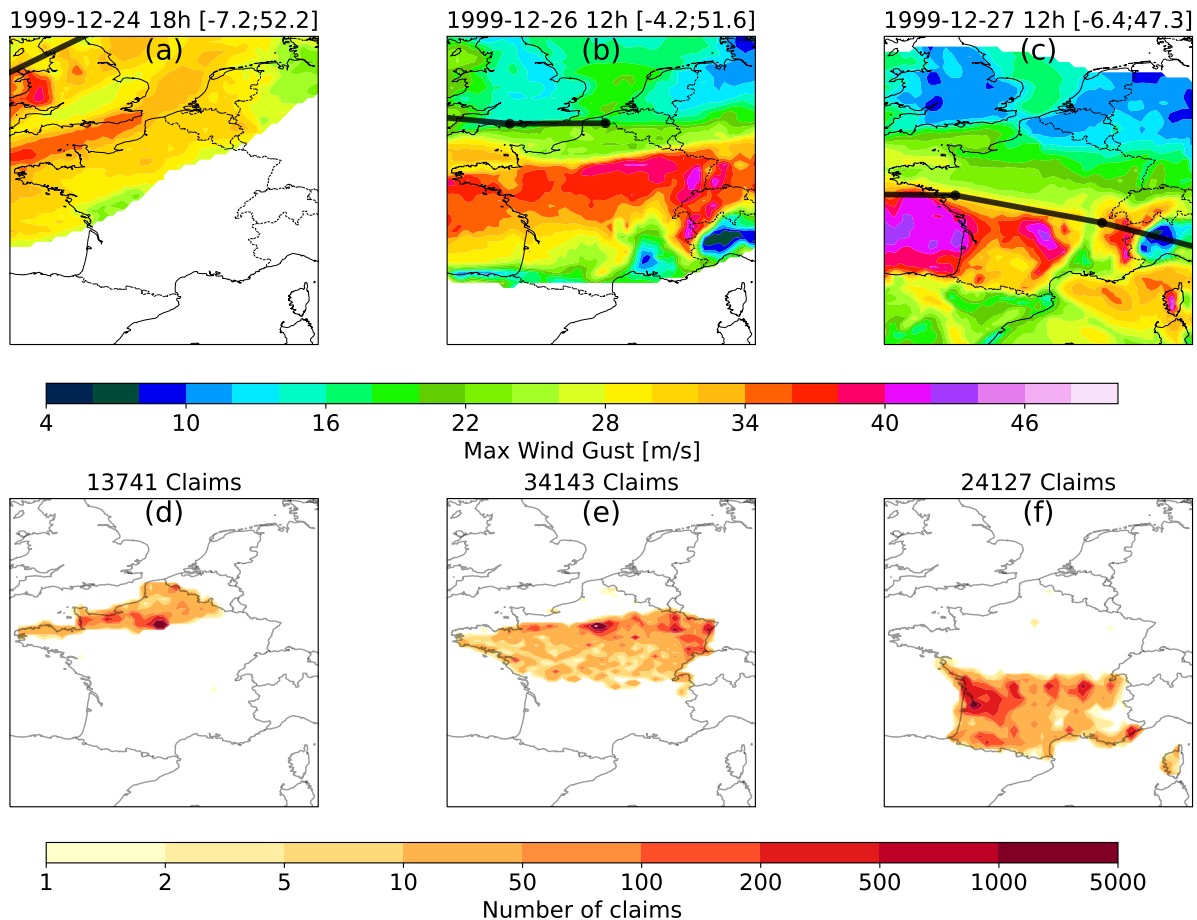

**Figure 8.** Association of claims to storms. Each column contains the maximum wind gust speed and the number of claims associated with a storm in Dec. 1999. Panels (a) and (d) show maps obtained for storm Anatol (*"1999-12-24 18h [-7.1;52.1]"*), (b) and (e) for storm Lothar (*"1999-12-26 12h [-4.2;51.6]"*) and (c) and (f) for storm Martin (*"1999-12-27 12h [-5.2;49.4]"*). Panels (a), (b) and (c) show wind-gust footprints and the storm trajectories (thick black lines), panels (d), (e) and (f) show the spatial distribution of the number of claims associated with each storm, while titles indicate the total number of claims for the whole event.

Storm Anatol, Lothar and Martin respectively landed on the $24th$, $26th$ and $27th$ December 1999. However, no claims were declared on the $24th$ and the impact of Lothar and Martin is likely mixed with the claims declared between the $26th$ and the $28th$. Restricting the analysis to claims reported strictly on the date of each storm would risk misattributing impacts.

Such misattribution can compromise the accuracy of vulnerability curves by linking damages to incorrect meteorological conditions. Furthermore, attributing claims to the wrong event and date may affect the proper aggregation of losses under the 96h reinsurance clause.




We applied our association method using the maximum gust wind speed of the three storms and the insurance claims for the period between Dec. 20 and Dec 30 1999 ($X_a = X_b = 3$ days). $approx 1514$ claims (out of 73605) cannot be attributed to any of the three storms. Fig.8 shows the footprints of storms Anatol, Lothar and Martin along with their associated claims. The distinction between their estimated impacts is particularly evident over central France and corresponds well to the shift of the highest wind gust. Our association procedure leads to a better focus on the impact regions of successive events, allowing us to distinguish their effects even when storms share a common impact area. Hence, the claim patterns Fig.7 can be refined to those in Fig.8.

We underline that the three storms have resulted in some damage in Paris. Although Storm Lothar had the highest windgust in the Paris area, the claim date played a key role in the distinction. This means that the claims observed in Paris and attributed to storm Anatol must have had a claim date earlier than December 23rd 1999 ($X_b = 3$ days before the landing date of storm Lothar). Similarly, the claims observed in Paris and attributed to storm Martin must have been declared later than $X_a = 3$ days after storm Lothar, so later than December 29 1999.

We highlight that damage is separated solely by the most intense wind gust. Our approach does not account for the persistence of strong wind gusts or the possibility that damage may result from the last storm, even if it did not produce the highest wind gust. This represents the primary limitation of the method. An alternative approach could involve using a different association method, such as linking events based on the closest track in terms of distance. However, for clustered storms affecting the same region, this may not be ideal. As illustrated in Fig. 8, weaker wind gusts can be observed near the centre of maximum vorticity. This suggests that some damage could be incorrectly attributed to areas with relatively low local wind gusts.

Although usual reinsurance gathers storm events with a 72 or 96h time difference, in case of major events such as Lothar and Martin, separation could be needed. The association also succeeds in capturing postponed damage that could be declared a few days after the storm event.

## 4.3 Case Study: Storm Klaus

The storm–claim association also underlines the relative contribution of smaller storms, which are usually discarded. As discussed in Section 2.1, it is crucial to account for all storms, as fast-moving systems or smaller depressions can generate strong surface winds. If these storms are not captured, all the damage may be attributed to the strongest storm, which might not be the one responsible for the winds at a given location. The example of Storm Klaus serves to illustrate the importance of including small depressions and accurately distinguishing their impacts.

Storm Klaus affected Southern France and Northern Iberia between January 23 and 24, 2009. It was characterised by an explosive deepening of 37 hPa within 24 h, which is relatively uncommon for this latitude (Liberato et al., 2011). This rapid intensification was driven by an extended and intense polar jet at upper levels. At the surface, strong wind gusts caused significant damage to infrastructure and forests (AIR, 2009).

This storm is often treated as a standalone event by many insurers. Claims filed within a 72 or 96 hour window around January 24, 2009, are typically attributed solely to Klaus. However, our association results reveal that Storm Klaus was part of a broader storm cluster. The tracking algorithm identified a preceding depression, which will be named Storm A, crossing



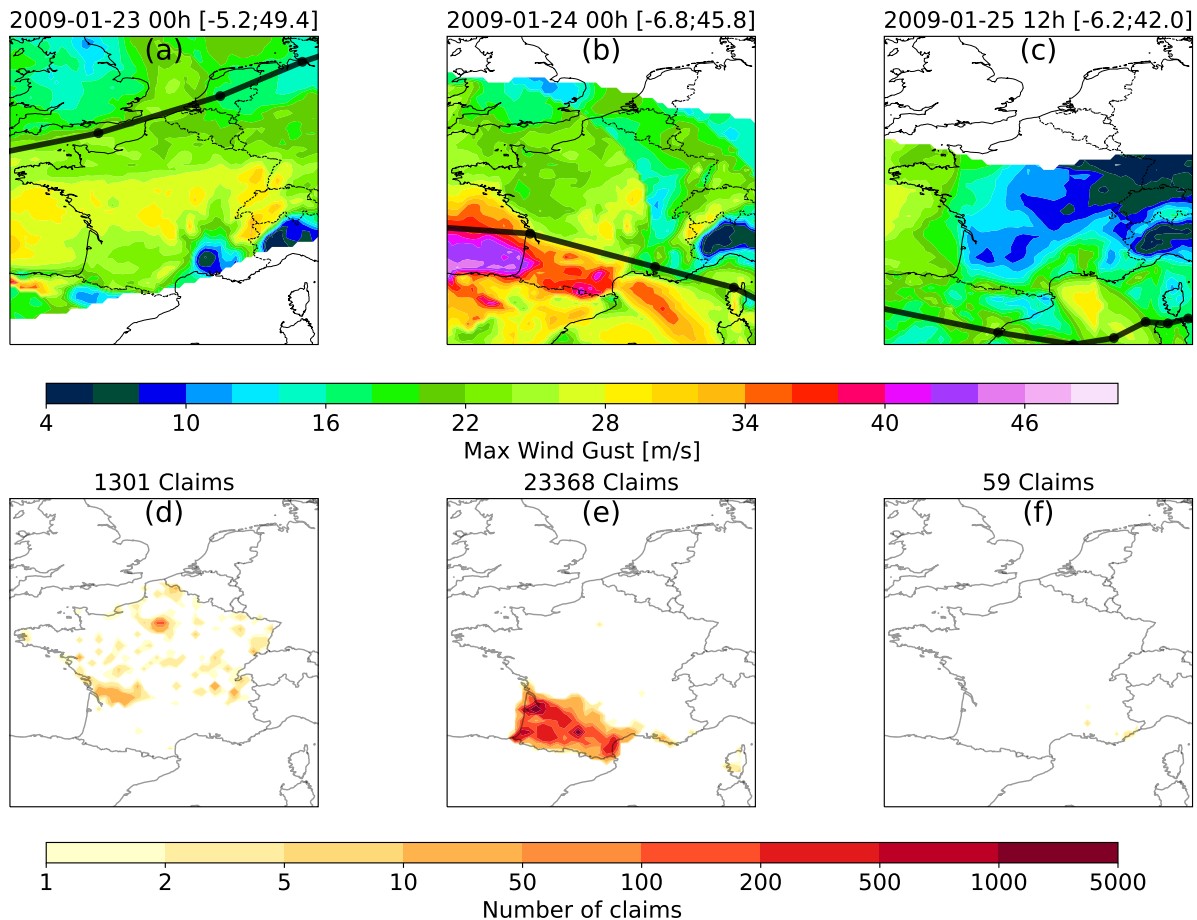

**Figure 9.** Same as 8 with storm A(*"2009-01-23 00h [-5.2;49.4]"*) in panels (a) and (d), storm Klaus (*"2009-01-24 00h [-6.8;45.8]"*) in panels (b) and (e) and storm B *"2009-01-25 12h [-6.2;42.0]"* in panels (c) and (f).

Northern France on January 23, 2009, which produced significant wind gusts exceeding $20\,\mathrm{m/s}$ in central and northern France. Additionally, a secondary low-pressure system developed on January 25, 2009, that we named Storm B, was potentially responsible for claims in the South-West of France. As illustrated in Fig.9, wind gusts generated by the January 23 and January 25 ETCs exceeded those generated by Klaus in several regions. This suggests that damages in northern and central France may not be attributable to Klaus alone. These findings demonstrate the added value of our storm-claim association method and highlight the importance of including smaller depressions in storm databases.

Misattribution of claims has several implications. Assigning all damages to Storm Klaus would overestimate its overall impact while underestimating the contribution of Storms A and B. Moreover, associating damages with Klaus's meteorological footprint implies weaker wind gusts than those responsible for the damage in some regions (see Fig.9). Such mismatches can degrade the accuracy of loss modelling and vulnerability estimation.





This example demonstrates the importance of the accurate storm–claim association, especially in cases of clustering. Attributing all impacts to Storm Klaus neglects the influence of smaller depressions. The relative contribution of these small systems within a 96-hour cumulative sum is essential for understanding the role of serial storm clustering. This study highlights potential pitfalls in the interpretation of clustered events, as aggregated losses, claims, or impacts over a period as large as 72h or 96h often encompass damage from multiple storm events. It remains unclear whether Klaus alone would have caused similar levels of damage had it not been preceded by another system, a consideration critical for understanding compound and cascading impacts.

## 5 Discussions and perspectives

Extracting quantitative impact information from insurance claims presents significant challenges. While such data provide a direct measure of socioeconomic impact, it is often influenced by variations in insurance exposure, both spatially and temporally. The raw nature of claim data introduces potential quality issues. Reported damages depend on the awareness of policyholders and claims managers, who may be influenced by external factors such as media coverage of the event. This leads to an under-representation of smaller or moderate windstorms, a phenomenon referred to as "historical perception bias" (Stucki et al., 2014). Additionally, the occurrence and severity of damage cannot be fully explained by physical storm variables alone, because socio-economic factors and inherent randomness also play roles (Birkmann et al., 2013). Careful post-processing was therefore essential to extract meaningful signals from claim databases.

This study introduces a robust methodology to associate high-resolution claims data with ETCs, enabling a more accurate attribution of damages. The method is based on three tunable parameters: a temporal window around storm occurrence ($X_b$ days before and $X_a$ after) and a minimum claim threshold ($n_{\text{claims}}$) to identify impactful events. Applied to windstorm claims from Generali France, the optimal window of three days before and after storm events and a gathering around storms with at least 50 claims effectively captures the storm-related damage signal. While the method was tested on the claim data from a single insurer in France, its design is adaptable to other perils, portfolios, and regions. The simplicity of the tuning parameters makes the method highly transposable to other outlooks. The cost function can also be modified to align more closely with different association perspectives.

The resulting claim–storm associations reveal coherent spatial structures and a strong correlation between local wind gust intensity and damage, validating the physical basis of the approach. The 40 most damaging events identified from 1998 match well with existing storm databases, and cost metrics per storm align with previous findings from the insurance literature (Frédération France Assureur, 2025; Mission Risques Naturels, 2021). This highlights the method's robustness and its potential for systematic storm impact identification.

This paper investigated the impact of clusters of storms using solely ERA5 reanalysis, which can be a limitation when focusing on historical storm events. Discrepancies between reanalysis and observational datasets in terms of wind gust values are well documented (Flynn et al., 2024). The proposed method is based not only on the date of the storm but also on the wind-gust value observed at the claim location. It can thus be expected that with different meteorological input data, different impacting



storms and consequently clusters could be identified. Exploring the sensitivity of outcomes to different meteorological sources
would be a valuable step in testing the method's robustness.

Although assumptions were necessary to attribute claims to storms, direct attribution remains inherently uncertain. In par-
ticular, it was assumed that the damage resulted from the storm associated with the highest wind gust. The whole method was
built to overcome the issue of not knowing the driver of the loss. As windgust is known to be one of the primary drivers of

damage, this variable was used to disaggregate claims associated with overlapping events. Incorporating alternative variables,
such as sustained 10m wind or upper-level wind, could lead to the detection of different impacting storms. However, recent
research shows that claim behaviour aligns similarly with various wind metrics (Fonseca Cerda et al., 2024). In the context
of storm clustering, attributing damage to a single event becomes especially challenging, as multiple storms with high wind
intensity are observed at the same moment. Overlapping storms with high wind intensities often act collectively, and damage

may arise from the persistent impact of several storms in quick succession. Because wind gusts are relatively localised, the
method can distinguish between storm events when high-intensity winds occur in distinct areas (e.g., Storm Klaus Sec. 4.3).
Conversely, in cases like Lothar and Martin (Sec. 4.2), where the same locations experienced high wind gusts from multiple
events, further analysis is required to compare the observed damage with what would be expected if each storm came alone.

Clusters of storms were defined from an impact-centric perspective, using Generali's claim data. The choice of a 96h win-

dow reflects Generali's reinsurance aggregation policy, but this duration could be adjusted depending on in-place reinsurance
policies. Larger aggregation windows could amplify clustering effects. While statistical metrics such as event dispersion could
also define clusters, they do not easily support event-based attribution as required by impact modelling.

Accurate attribution has direct implications for reinsurance. Most contracts aggregate losses within a fixed time win-
dow—typically 72h hours, but extended to 96h hours in Generali's case. As the time window increases, closely spaced events

are more likely to be grouped, reducing the number of reimbursable events and leading to lower overall reinsurance costs.
Modifying this window can affect loss estimates and the financial response of reinsurers. We also observed that claims are
sometimes reported up to three days before or after the actual storm date. Reassigning these claims to the appropriate storm
would allow for a broader temporal grouping of losses. In practice, a 96h reinsurance window could then encompass claims
occurring over a total span of up to 240h.

Understanding historical storm clustering and comparing these patterns to those simulated in catastrophe models is essen-
tial for refining reinsurance structures and improving financial preparedness (Kaas, 2009; Khare et al., 2015). Clustering also
raises questions about evolving vulnerability. Current catastrophe models often assume static vulnerability, failing to account
for changing exposure or structural fatigue due to successive events. For instance, a building weakened by one storm might
be more susceptible to damage from a subsequent event. Most vulnerability functions developed in the literature assume event

independence (Dorland et al., 1999; Klawa and Ulbrich, 2003; Heneka et al., 2006; Schwierz et al., 2010; Prahl et al., 2015;
Pardowitz et al., 2016). These assumptions should be critically revisited to assess how vulnerability evolves under compound
stress. The correct attribution of losses, as proposed in this paper, is an important step toward building more dynamic vulnerabil-
ity curves. Precise knowledge of the physical conditions leading to loss is essential to disentangle meteorological drivers from



structural susceptibility. Future research could investigate whether the same building characteristics are consistently vulnerable during storm clusters or whether these features exhibit context-dependent behaviour.

A key strength of this work is its capacity to disentangle and attribute damage within storm clusters. Case studies, including storms Anatol, Lothar and Martin (Sec4.2), as well as storm Klaus (Sec4.3), demonstrate how simple date-based attribution is insufficient to isolate the impact of temporally or spatially overlapping events. Our study reveals that the most damaging storms are frequently part of storm clusters, confirming earlier work by Vitolo et al. (2009), which highlighted the tendency of intense ETCs to occur in series. This analysis further shows that the total loss associated with a cluster event is distributed across multiple storm members. This highlights the risk posed by storm clustering, where the overall damage arises from the combined effects of several interconnected storms, rather than being attributable to a single event.

## 6 Conclusions

The association method presented here offers a valuable framework for improving our understanding of storm-related damage. We underline that a temporal window of a few days before and after, as well as a threshold over the minimal number of claims per event, was essential to capture all the impact. This method enhances the attribution of damage to specific physical events, which is key for risk assessment, loss estimation and prevention strategies. It also supports the identification of storm clusters and provides a foundation for assessing compounded risk. The study underlines an exacerbated impact linked to clusters of storms which are responsible for $85\%$ of the total losses since 1999 for Generali. We also showed that storms associated with damage exhibit more clustering than the set of potential ETCs impacting France. Within these clusters, the percentage of the loss held by the most costly storm varies widely, with an average contribution of $70\%$ of total losses. Notably, this share is not influenced by the storm's order of arrival. Case studies of well-known high-impact storms further validated the method's ability to disentangle the damages caused by successive storms. These findings represent a step forward in understanding how the combination of hazard characteristics and exposure dynamics contributes to storm-related losses and support more nuanced approaches to managing compound weather risks.

## Appendix A: Sensitivity of the cost function

The cost function $f_{cost}$ defined in section 3.2 varies as a function of the weight assigned to the frequency metric ($w_{freq}$). Fig. A1 shows the optimal values of the parameters $X_b$, $X_a$ and $n_{\mathrm{claims}}$ as a function of the varying weight. It can be underlined that the optimal values of all the parameters are identical for the weight varying between $0.3$ and $0.6$. This means that, with a balanced penalty between the precision and frequency metrics, the optimal parameters are identical.




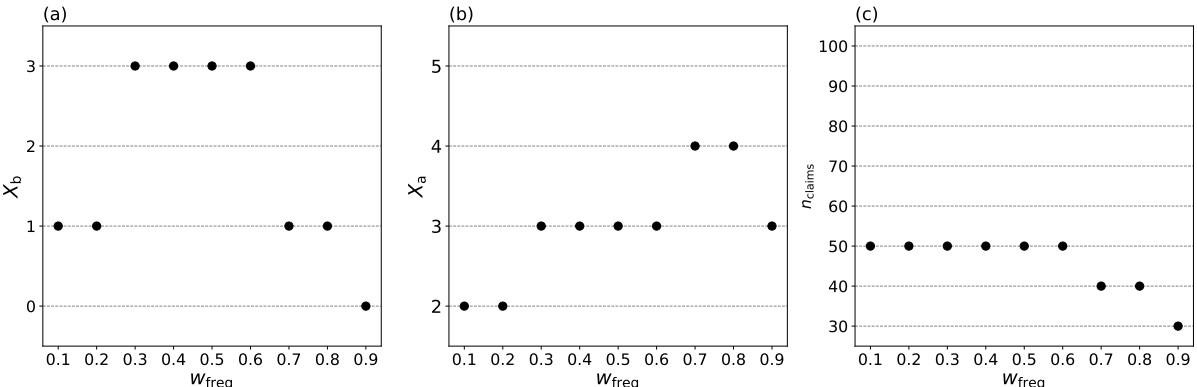

**Figure A1.** Optimal value of $X_b$ (a), $X_a$ (b) and $n_{\mathrm{claims}}$ (c) as a function of the weight of the frequency metric $w_{freq}$

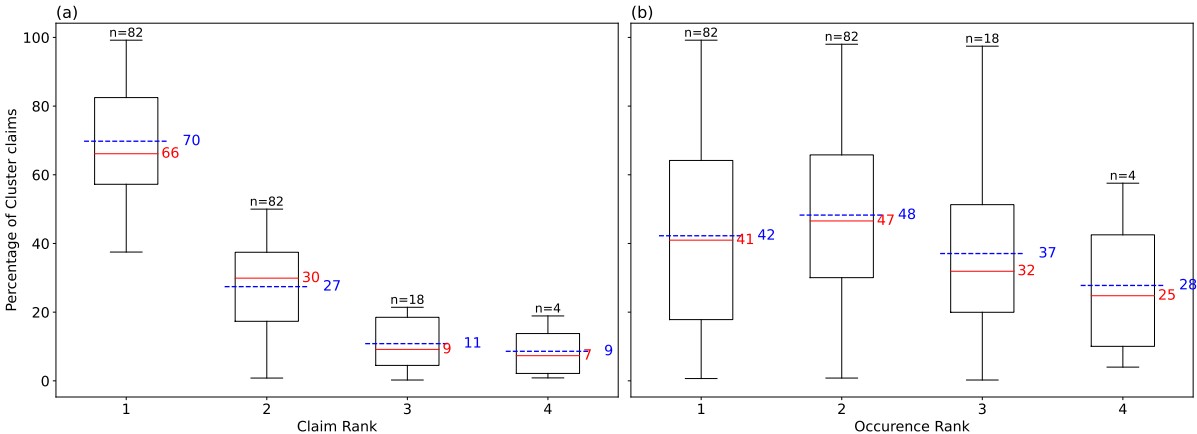

**Figure B1.** Distribution of the number of claims within the members of the cluster as a function of the claim rank (a) and occurrence rank (b). Thick red lines indicate the median, and blue dashed ones the mean. The numbers above each box correspond to the number of members used for each box plot.



## Appendix B: Results

### B1 Distribution of claim intensity within clusters

Fig. B1 shows the share of the claim intensity as a function of the claim and occurrence rank. A claims rank of 1 corresponds to the member of the cluster with the most number of claims.

*Code availability.* Scripts to reproduce the main results of this publication are available a thttps://doi.org/10.5281/zenodo.15771837 (Hasbini, 2025)

*Data availability.* ETC tracks were provided by Matthew Priestley. ERA5 data is openly available in Copernicus Climate Change Service Climate Data Store at https://doi.org/10.24381/cds.bd0915c6. Generali claims datasets analysed in the current study are not publicly available as they are proprietary to the company.

*Author contributions.* LH, PY, LB and AP conceptualised the experiments. LH produced the numerical experiments and analyses. LH and PY contributed to writing the paper.

*Competing interests.* The contact author has declared that neither of the authors has any competing interests. Authors LH, LB and AP are employed by Generali France.

*Acknowledgements.* This work was supported by the French ANRT who funded LH's PhD thesis. We thank M. Priestley (U Exeter, UK) for
useful discussions on the storm tracking algorithm.



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
