# Peer review of "Unravelling the wind impact of clusters of storms, a case study over the French insurer Generali"

_EGUsphere, 2025_

## Referee Comment (RC3)

**Unravelling the wind impact of clusters of storms, a case study over the French insurer Generali**
**Egusphere-2025-3138**

This paper aims to (i) link insurance claims to individual ETCs, (ii) apply the method to clusters of ETCs and (iii) assess the impact of storm clusters on insured losses. Overall, the paper achieves the aims. In places the language used in the paper is overly emphatic for a piece of scientific writing and rewriting is needed to ensure that all statements are supported by evidence (see comments below for specific examples).

**General comments**

1. In many places the authors describe their work as fundamental, significant, valuable and a step forward. It should be left to the reader to evaluate the importance and novelty of the results presented. Please remove all emphatic adjectives.
2. There are several statements in the paper referring to small storms, however I could not find any description of how storm size was measured. The authors need to include this, or be more precise about what is meant by small storms, perhaps it refers to the intensity or duration of the storms rather than their physical dimensions?
3. The evidence to demonstrate the validity method is described in sections 4.2 and 4.3. This is after results using the method to link storm occurrence to losses (figure 6b, section 4.1). It would make more sense if the paper were reordered to demonstrate the method before it is used. Also, it would be helpful to provide quantitative information regarding the partitioning of losses between the storms in the cluster as well as partitioning the number of claims.

**Specific comments**

4. **Title**: Is the word 'Unravelling' necessary in the title? This verbose language is common when using AI to suggest a title so should be avoided.
5. **Line 5:** The authors describe their own work as a cornerstone for insurance and reinsurance processes. This is a very bold statement and should be left to the reader to determine how fundamental the methodology presented is.
6. **Line 19:** Here and elsewhere the storms are described as being displaced, I think path or track might be a more familiar word to use.
7. **Line 24:** The SSI should be explained in more detail. What does it measure?
8. **Line 32:** A Poisson distribution describes events that happen randomly and independently. I think that there is no reason why a Poisson distribution with a large mean cannot also have a large variance.
9. **Line 44:** The dispersion metric can be used globally I believe. Although I agree that it might not be suitable for impact assessments.
10. **Line 55:** How does the representation of hazards in insurance loss models lead to damage prevention?
11. **Line 77:** This sentence does not have an end. What advantages are unlocked? Perhaps 'unlocking several advantages' is not needed in this sentence?
12. **Line 79:** Where in the paper are the environmental factors leading to damage investigated?
13. **Line 79:** Here the authors describe their work as 'significantly enhancing our understanding of storm clustering events'. This is a bold claim and perhaps best left to the reader to decide on the usefulness of the paper.
14. **Line 83:** The assessment of reliability is difficult since there is no truth with which to evaluate the method. Perhaps remove the word reliably?
15. **Line 98:** By displacement of trajectories, are the authors referring to the minimum storm track length?
16. **Line 101:** What is the impact distant from? The storm centre or metropolitan France?
17. **Line 105:** Closest point to what? Does this refer to the location along the storm track that occurs closest to a longitude of 7.5W?
18. **Line 110:** It is not clear what the numbers in the curly brackets refer to.

19. **Line 113**: Earlier it is stated that a cyclone duration of 24hours is used, but here the authors state that the conditions on cyclone duration and intensity are 'relaxed'. What does this mean?
20. **Lines 114 and 252**: How do 'relaxed' constraints on cyclone duration and intensity refer to the speed and size of the storms considered? Additionally, on line 252 the authors refer to small storms, how is storm size determined?
21. **Line 118**: I don't think the authors have included any evidence to support their statement that including storms 'of all scales' reduces the bias. What bias are they referring to and how have they determined the scale of the storms?
22. **Line 124**: Earlier the radius of the storms considered for impact was set to 1300km but here it is reduced to 700km. Which is used in the study and why are 2 radii for impact mentioned?
23. **Line 177**: How are the robustness and reliability of the results quantified?
24. **Line 187**: What does the closest local maximum refer to?
25. **Line 188**: What does the number of local maxima refer to?
26. **Line 201**: Here the authors state that lighter colours indicate better results. What is meant by better and is this the case for all 3 tuneable parameters?
27. **Line 237**: How are the authors quantifying the accuracy/preciseness of their method? What are they comparing the method to?
28. **Line 248**: Here and in 12 other locations the authors use the phrase 'underlines'. This is quite repetitive use of language and alternative words could be used.
29. **Line 259**: Figure 4 does not show the intensity of storms or the vulnerability, so it is not clear what the authors are using to support this statement.
30. **Line 263**: Shift usually means a change. I believe the authors are simply referring to a difference here.
31. **Line 266**: What does 'cst' mean? Is this euro cents?
32. **Line 268**: What is restricted about the winter 2018/2019 analysed by Mision Risques Naturels?
33. **Line 276:** How is the impact of the clusters measured? What criteria is used to class clusters as high-impact?
34. **Line 277:** Is an impacting storm one that resulted in loses for Generali only or one which has a track within 1300km of France?
35. **Line 302:** What is meant by 'far from the 100%'?
36. **Lines 312-324:** These paragraphs repeat what is describes in earlier paragraphs so should be removed.
37. **Lines 397-403:** This paragraph is speculative. It does not describe the results from the paper so should be reworded or removed.
38. **Line 417:** What is meant by a gathering around storms with at least 50 claims?
39. **Line 419:** What is meant by 'highly transposable to other outlooks'? I am unsure what an outlook is.
40. **Line 443:** By 'came alone' are the authors referring to non-clustered storm events?
41. **Line 474:** The authors evaluate their framework as valuable. This evaluation should be left to the reader.
42. **Line 484:** The authors describe their findings are a step forward in understanding. This evaluation should be left to the reader.

**Typographical errors**

1. **Line 23:** The sentences describing windstorm metrics should form a separate paragraph since it introduces a new idea to the sentences preceding them.
2. **Line 47:** Do you need both event and impact here?
3. **Line 65:** Here 'including' should be 'such as'.
4. **Line 89:** I think 'over' should be 'including'?
5. **Line 93:** ERA stands for ECMWF ReAnalysis so it is not necessary to include the words 'historical reanalysis' afterwards.
6. **Line 103:** I think 'landfall' is more commonly used than 'landing'.
7. **Line 127**: Should 70km be 700km to be consistent with line 124?
8. **Line 149**: The 'st' in italics should be roman text.
9. **Line 203**: What is the 'peal date'?
10. **Line 354:** Why is approx. in italics?

**11. Line 426:** Why is the word 'soley' included in this sentence?
**12. Line 449:** Why is an 'double hyphen used here?

---

## Author Comment (AC1)

Broadly I like this paper. The findings may be a bit specific to this work and I wonder about the scope for broader applicability but its a nice piece of work which solves a clear problem in working with loss data.

We thank the reviewer for the developed comments and the interesting feedback. The responses to each comment are in blue.

My more specific comments are therefore mostly minor:

17. This is a global figure? Seems very high for Europe, even if Economic rather than Insured loss.

Indeed, different estimates can be found. We will rephrase this sentence to:
"In Europe, they rank among the costliest natural hazards with billions of economic and insured losses per winter."

25-29. I Don't entirely understand what this bit is saying.

The SSI metrics is defined using the wind (or wind-gust) and a percentile of this field (usually 98$^{th}$ percentile). The simplicity of its formula makes it easy to compute over land at any given day. However, this metric does not consider the lifecycle of the storm. In particular the duration of the storm or its deepening ratio might be other important variables to consider when trying to understand storm damage. Also, the presence of consecutive storms, which cannot be integrated directly in the SSI, is likely to influence the damage. We will rephrase this section to make it clearer.

45-49. Seems well justified.

Figure 1 caption - Should 'different by' be 'defined by'?

Thank you for spotting this, we will change it.

120-130 - Interesting discussion of clustering metrics and comments on how a clear definition has not been fixed, or at least not used within the insurance industry. From a loss perspective is it useful to split two clusters affecting France at the same time into separate clusters if they do not overlap?

This depends on the reinsurance policies in place in the company. Usually it is more advantageous to gather the expenses over one event as this will correspond to one request to reinsurance; but in some cases, like Lothar and Martin (in Dec. 1999), when both storms resulted in important damage, splitting the two events was more advantageous. Consequently, the ability to split or gather the events depending on the reinsurance contracts in place represents a great advantage compared to the initial situation, where events cannot be distinguished and are automatically grouped.

155-159 Clearly a good dataset with locations recorded. Postcode level is surely fine for ERA5 resolution hazard.

Thank you. Yes: post-code level is already a good resolution when comparing with ERA5.

164 - Good observation and an important problem to address.

174 - I can believe this but can you give a bit more explanation/justification. It reads slightly like you argue that the loss data over-represents big storms then remove the little storms yourself. From figure 2a I interpret that lowering N leads to more claims dates than storm dates which is a different problem.

The loss data is indeed biased around major storms because of public perception and media covering. This means that when filling the claim date, the insured person might be more likely to fill a date corresponding to a well-known storm event. This is corrected by the use of storm footprints, defined on their tracks and wind gust value. Using this data, the claims are associated to the storm with the highest wind gusts, which is not necessary the storm which raised the most attention (in the case of consecutive storms for example).

Another potential source of error occurs when the insured person does not know the exact date of the damage (for example, during holidays; when s.he is away). In such cases, the damage can be filled over any day. We became aware of this limitation by spotting some days with less than 10 claims across the entire France. Given the spatial extent of winter storms, such low claim counts are highly unlikely. Based on this, we adjusted the data to group claims into "reasonably large events." For example, Generali owns approximately 1million contracts over France, which represent 3% of the market share in France.

We will clarify this in the related section.

203 - 'Enables the capture'

We will fix this mistake.

Section 2 - Overall I find this interesting. Loss data is often focused around extreme events for vulnerability development and this study is an opportunity to work with a sufficiently complete loss history that the impact of smaller events is observed.

This is exactly the point we want to raise. In the case of winter storms the succession of smaller events can sometimes be as costly as the occurrence of a major storm.

246 - How typical Generali's portfolio is of the French market is crucial here. I expect the spread of risks cannot be shown but some commentary could be made as to whether the missing events were very focused on a particular region etc. Some commentary on the exposure comes in 271.

It is not entirely clear which storm Météo France refers to when mentioning the event of 16–17 December 2019. The associated damage is likely captured in the storms we identified on 20 and 21 December (see figure 3 of the original manuscript).

The January 2012 period, during which storm Andrea occurred, was marked by a succession of storms. Andrea mainly affected northern France, where Generali has exposure. In our database, three storms from January are linked to claims. The resulting damage is distributed across these events, which likely explains why Andrea does not appear among the 40 costliest storms.

Storm Calvann, which impacted western France on 2 January 2003, also appears in our database but is not listed among the costliest events. This is most likely because Generali has fewer policies in that region.

We will clarify these points in the revised version of the manuscript.

Fig 4 - Is there a relationship between total storm count and cluster counts? The grey bars corresponding to less clustered years appear to group in the 2000s which is also a period of low overall activity based on the red dots.

As the counts are made over a restricted period (the winter months), more storms over a fixed window means higher chances of clustering. From a statistical perspective, the storms are in fact

more likely to the be close to an existing storm date.
In fact, over the 2000s', Generali experienced few storms causing damage.

Section 4.1 seems like the most crucial results, using the dataset and methods discussed in much of the rest of the paper to draw conclusions.

We will move this section at the end of the discussion section (actual position of section 4.3) for clarity.

Fig 5. I would expect b) cost per claim to be similar, as observed. But do a) and c) depend on the definition of a storm? If your catalogue contains more small storms the total cost and total claims would be lower, as is observed.

This is one of the main results of the method. Including smaller storms result in lower values for the total cost and total number of claims per storms, compared to catalogs containing only intense events.

Fig 6. Could we see the spread of losses within clusters for different cluster lengths? For example in a 3 storm cluster is the second ranked loss still 26%?

[Figure]

*Figure 1*
*Distribution of the loss within the members of the cluster of at least 3 storms as a function of the loss rank. Thick red lines indicate the median, and blue dashed ones the mean.*

We selected the clusters containing at least 3 impacting storms. For each storm member of a cluster, we compute its share of the loss as the total cost of the member divided by the total cost of the cluster. A loss rank of 1 corresponds to the costliest member of the cluster. Figure 1 above shows the percentage of the cluster loss as a function of the loss rank.

As we observe only 18 clusters of this type, statistics might be less robust. We can draw similar conclusions to the ones of the manuscript using the full set of clusters events (figure 6a of the manuscript). The 2[nd] costliest member of the cluster still represents, on average, 26% of the total losses of the cluster.

Section 4.2 shows some value to the method. Often losses from these storms are indistinguishable.

Fig 7/8 what is the resolution of the loss contour maps? The scales go to 6k and 5k respectively but within what area are 5k claims observed? Also there are some white dots that look like they hit Paris that don't seem to be explained.

The resolution is the same as the one of ERA5 (0.25°x0.25°). The claims, available at building level are aggregated over this grid. The white spot in Paris corresponded to an issue in the color bar ticks. The issue is fixed in the new version of the figures (see below).

[Figure]

*Figure 2 - Association of claims to storms. Each column contains the maximum wind gust speed and the number of claims associated with a storm in Dec. 1999. Panels (a) and (d) show maps obtained for storm Anatol ("1999-12-24 18h [-7.1;52.1]", (b) and (e) for storm Lothar ("1999-12-26 12h [-4.2;51.6]") and (c) and (f) for storm Martin ("1999-12-27 12h [-5.2;49.4]"). Panels (a), (b) and (c) show wind-gust footprints and the storm trajectories (thick black lines), panels (d), (e) and (f) show the spatial distribution of the number of claims associated with each storm, while titles indicate the total number of claims for the whole event.*

[Figure]

*Figure 3- Same as 8 with storm A("2009-01-23 00h [-5.2;49.4]") in panels (a) and (d), storm Klaus ("2009-01-24 00h [-6.8;45.8]") in panels (b) and (e) and storm B "2009-01-25 12h [-6.2;42.0]" in panels (c) and (f).*

365-368 Losses are often due to debris, hence consecutive storms may impact losses. This is very difficult for the insurance industry to study and datasets such as this may enable that.

Thank you for raising this. We will clarify this point in the text. The dataset could indeed identify the additional impact related to successive storm as it can isolate the impact of individual storms.

Section 4.3 is also interesting. As stated in 384 Klaus is often treated as a single event.

Thank you for the feedback.

---

## Author Comment (AC2)

I read with great interest the paper by Hasbini et al. and found that it provides useful and insightful links between scientific research and industrial applications, particularly in the (re-)insurance sector. Nevertheless, I have several queries, especially regarding the presentation, methodology, and motivation. More specifically, I found it difficult to fully understand the methodological approach, and I have a few concerns about the procedure used to attribute impacts to unique storms. Despite these issues, I find the content very interesting and would therefore recommend a major revision.

Since my concerns mainly relate to the methodological approach, in the following, I focus primarily on the first half of the paper. Addressing these comments could potentially lead to changes in the results or, at the very least, contribute to a better understanding of the conclusions.

We thank the referee for the review of our manuscript. We also thank the reviewer for the interesting discussion, which emphasizes the potential limitations and challenges of our analysis. The individual answers to each of the comments are in blue in the rest of the document.

1. Presentation

I found the use of language generally clear; however, I had some difficulty following the text overall. In my opinion, the introduction lacks a consistent narrative and does not make sufficiently meaningful use of technical language. I would strongly suggest revising it according to the comments below:

Line 21: Although I agree that tracking methods tend to show higher agreement on the tracks of well-developed storms, I would argue that tracking algorithms can still diverge in their results even for the most intense cyclones. Including a relatively brief review of tracking-method intercomparison studies would strengthen this point.

Indeed, the choice of the tracking method can highly condition the results (as highlighted later in the review). We will rephrase the section to refer more specifically to the studies on tracking-method intercomparison of (Flaounas et al., 2023; Neu et al., 2013).

Lines 27–29: The terms "explosive cyclogenesis" and "strong jet stream" cannot be straightforwardly characterized as physical characteristics. Please revise this phrasing. Additionally, the purpose of these phrases is unclear. In fact, the second half of this paragraph appears to consist of loosely connected statements rather than a coherent argument. I suggest restructuring it for clarity. Probably discussing cyclone dynamics is not of strong concern for this paper.

We included this section to give a dynamical background for damaging storms. Indeed, we do not specifically treat cyclone dynamics in the rest of the paper. We will clarify this section to refer more clearly to the dynamical mechanisms.

Lines 30–31: The type of distribution may indeed vary depending on the tracking method used. However, I am not sure I understand how the distribution type justifies the varying number of ETCs "observed at a given location and for a given period." Please clarify.

We will clarify this sentence.

Lines 33–34: The statement is vague. Please be more precise.

The definition of serial clustering given by (Dacre & Pinto, 2020) is deliberately vague, as no unique definition of the clusters exist. Following this, the clustering used in this paper was defined as an

abnormal frequency of storms over a restricted time period. This period was fixed to 96hours, in order to align with the length of reinsurance events. We will clarify this section to emphasize more clearly that no unique definition of cyclone clustering exists, and we chose a "user-based" definition.

Line 34: RWB typically occurs due to external forcing on the waveguide — for example, from the outflow of warm conveyor belts. I assume the authors intended to express a different idea here. Please revise, clarity or omit.

We will revise this in order to refer more precisely to the arguments raised by (Pinto et al., 2014; Priestley et al., 2017).

Lines 38–40: Dominate what? Which "dynamics" are being referred to? It seems that the term dynamics is being used as a generalization for "all the above." Please be more specific and precise when using scientific terminology.

In this sentence, "dynamics" refers specifically to the secondary cyclogenesis. We will clarify this section to refer more specifically to the mechanisms involved. The sentence will be rephrased as follows: "Recent studies (e.g. Pinto et al., 2014; Priestley et al., 2020) have demonstrated that cyclones formed by secondary cyclogenesis are more numerous during clustered periods over Western Europe. Better understanding secondary cyclogenesis formation is therefore key for improving our knowledge of clustering mechanisms. Such analyses are beyond the scope of the present study."

Lines 46–48: Why is a 4-day window necessary? Please provide justification for this choice (seems that this is done in the discussions section and thus it comes too late).

The choice of temporal windows depends on the research or practical objectives. From an event and impact perspective, an absolute definition with a 96-hour window, corresponding to the length of reinsurance contracts for Generali, is particularly relevant for capturing the temporal clustering of extreme events. We will add this justification earlier in the text.

Lines 49–53: This section seems more like an introductory paragraph to clustering. Consider integrating it more clearly with the broader discussion.

In this section we want to highlight that understanding cyclone clustering is of great interest not only because of the physical perspective but also from the impact perspective, as seasons with intense clustering were also association with substantial losses. We will rephrase this to make the connection with the rest of the section more direct.

Line 61: The meaning of "driving characteristics" is unclear. Please define the term more precisely.

In this section, we emphasize that characterizing storm events, solely by aggregated surface wind is too simplistic to determine whether their physical characteristics increase the likelihood of damage. Such an approach overlooks important aspects of storm evolution (e.g., secondary cyclogenesis, explosive deepening, successive events) that can strongly influence both the occurrence and severity of damage. We will clarify this point more explicitly and describe more clearly what "driving characteristics" refers to.

Lines 62–63: I agree with the general point, but in some cases, a strict one-to-one attribution of impacts to individual storms is difficult, if not impossible. For example, if a new storm develops within the frontal region of a mature storm, then attributing damages exclusively to one or the other becomes completely arbitrary.

We agree with this point. However, from an insurance and reimbursement perspective, claims still need to be grouped around specific storm event.

What we also aim to highlight in this paper is that directly attributing damage to a single storm is often complex. As you mentioned, the development of successive storms over frontal zones can further complicate attribution. One objective of the manuscript is also to understand how frequently such situations occur, particularly when it is unclear which storm was the primary driver (for instance during clustered episodes. This will be discussed in the revised manuscript.

Lines 83–85: Reading the objectives, it seems that resolution-related issues are not thoroughly discussed earlier in the text. If necessary please do so. Additionally, while you mention the limitations of aggregated wind speed data in attributing impacts to a single storm, the importance of distinguishing between storms is not sufficiently developed (see also comments on methodological approach). I overall recommend formulating the introduction to better align it with the study's stated objectives.

In fact, the resolution-related issues are more a discussion point, as we did not test results obtained with wind-speed of different resolutions. We will reformulate the first research question as follow: "How can insurance claims data be reliably linked to ETC?".
We will also highlight in the introduction and in the discussion about damage dataset, that aggregated losses (at country and department levels) make it impossible to distinguish between successive events.

We will consider the limitations you raised and rephrase part of the introduction to make it align better with the results presented in the rest of the manuscript.

2a. Storm tracks approach

My impression is that cyclone tracking is naively taken for granted considering every track as "a storm". It is quite clear from many cyclone tracking papers that the number of "storms" can be easily tuned and lead to much different results. In the case of this paper, with some very simplistic calculations and supposing a rather constant number of storms per year, then the authors seem to conclude to 100 storms per 6-month season (4439 storms for 45 years), i.e. ~16 storms per month, or more naively, one storm every other day close to France. The potential conclusion of the important role of clusters in provoking damages is thus "predecided" by the storm tracking approach. I feel that the methodological approach should be better discussed and the reader should be provided with more insights about the tracked features (see specific comments below).

[Figure]

Figure 1- Distribution of storm tracks with an impact radius of 700km for the set of all storms (a), storms resulting in impact for Generali (b) and storm resulting in clustered impact (c)

[Figure]

*Figure 2- Same as Figure 1 with an impact radius of 1300km*

Indeed, this could be interpreted as a large number but all the atmospheric depressions are, at this stage, the only potential candidates for storm impact. As mentioned later, only a few tracked depressions have led to some impact for the insurance company. Most of the storms counted in this climatology are either too far from France, or not deep enough and not associated with intense wind.

The reason for which we kept all the depressions in the initial set of tracks is because we realized that adding more constraints to the tracking algorithms (such as increasing the threshold over minimal lifetime, or decreasing the radius of 1300km) would discard some major storms with impacts. With a smaller radius, some important wind gust associated to storm Lothar were discarded. Also, some storms occurring at the beginning of January 2018, which were known to be associated with important impact (Vautard et al., 2019), were not present in the set of tracks. Therefore, we acknowledge that trial-and-error tuning on observed and documented events has to be performed to achieve a balance between the number of tracked storms and their potential impacts.

Figures 1 and 2 represent the number of storms detected with impact areas of respectively 700 and 1300km. We see in these two illustrations that the number of storms is drastically reduced after filtering over storms associated with impact for Generali. The more exposed regions are the North of France and Brittany, known to be particularly exposed to storm events, and for which we find on average 3 to 4 storms per winter, with a radius of 700km, or 6 with a radius of 1300km. The number of storms associated with impact for Generali per winter as well as the most exposed areas is consistent with the two selected radii.

Specific comments

Line 109. The storms seem to be treated as "all of the same". For instance, the same radius of influence is used (1300 km) for all tracked features, but I would argue that this is a rather unrealistic assumption. From the perspectives of a morphological approach, the impacts are expected to take place close to the center and along the fronts. So a circular area with a radius of 1300 km seems to be overwhelming. Maybe not so overwhelming for capturing the extent of the fronts but for a small storm this would be certainly the case. This radius actually compares "rather too big" with the surface of France. This seems to further favor the collocation of storms and the conclusion of clusters' high importance. I would advise the authors either to use a dynamically varying effective area for the storms, e.g. change according to intensity, or better, to detect and attribute specific wind patterns to storm track points.

Thank you for the suggestion: better characterizing the footprints could indeed improve the method. The radius was chosen at 1300km following some literature using radius of 12° (Hawcroft et al., 2012;

Sinclair & Catto, 2023). Smaller radius of 6° can also be found in the literature (Gramcianinov et al., 2020; Zappa et al., 2013) but some case studies over our set of identified storms reveals that this radius was not sufficient to capture all the wind-related impacts.

We believe automatic association of wind to frontal systems and storms tracks would be an interesting research project, but this is beyond the scope of our paper. As the aim of the proposed manuscript was more to get the physical insight from an impact dataset, we decided not to include it. Nonetheless, adding frontal detection to this approach could be a really interesting perspective as it would also give insight on the structure of fronts most likely to result in damage.

Finally, extensive literature also showed that applying spatial radius filtering over storm tracks is usually enough to "decontaminate" the footprints (Copernicus C3S, 2025; Lockwood et al., 2022).

Line 96. I am not sure what is meant by "potential noise". As far as I know, the TRACK algorithm identifies and tracks local maxima of relative vorticity. Relative vorticity is a high frequency field. Even a relatively coarse resolution of 0.25x0.25 would pose a computational challenge due to numerous identified local maxima that need to be tracked in time. So smoothing is necessary (I would actually argue that noise is not even appropriate here). How much smoothing is applied and what is the native horizontal resolution of the input field matters greatly for the eventual number of tracked features. These aspects, plus the arbitrary choice of cyclones duration (line 101), are all important to conclude to a high number of tracks. My impression is that in the trascks dataset might be included features (i.e. local maxima of vorticity) that could be rather small, or nested within greater cyclonic structures (e.g. local maxima of vorticity nested in a cold front). Therefore, several (many?) of the tracked features could be hardly considered as "distinct storms".

This is not a critical comment, because bottom line, there is no right or wrong when it comes to the number of tracks, albeit the reader needs to have a good statistical knowledge of the characteristics and nature of tracked features. For instance, it is not rare for tracking algorithms to perceive a "large scale cyclone" with 2 or 3 distinct centers as a set of that many different storm tracks. Whether there is one "big" cyclone or 2-3 "smaller" ones is an arbitrary choice and depends on the tuning of the cyclone tracking method. However, if there are large heterogeneities in the intensities of tracked cyclones, or if many of these cyclones overlap in time, then one needs to adjust their approach when attributing specific impacts to "storms" (e.g. adjusting the radius of influence). Lines 113-115 suggest that cyclone tracking approach was designed to include heterogeneous storms. This needs to be shown, discussed and linked with an accordingly adjusted attribution methodology (see also below).

We can justify our method as follow:

- Filtering for intense tracks is essential when working with Eulerian statistics or aiming to derive global measures. In contrast, for an event-based approach, the absence of the tracks (whether unique or multiple) of the event in question becomes a major limitation. While diving into such analysis, we realized that the usual storm criteria can be overly, excluding certain some high-impact storms (for example storms of January 2018).
- The potential division of one "big" cyclone into smaller subset is a particular case of interest. With the choice of the storm tracks we could expect this to happen. On top of this the association method might also associate the damage to each "member" of this bigger depression. And as you underscore, we would call a cluster this big depression which was divided into 2 to 3 smaller storms. The main concerns are then: "Can we really call a cluster of storms an event for which the several centers where part of bigger system?", "Shouldn't they be combined ?"
We argue that the more we are able to specify and divide the loss, the better it is. Studying

the type of objects encompassed in this cluster definition and whether they were subdivisions of the same global represent correspond to a different research question.

- Finally, we emphasize that storms are also filtered through the association process, which is specifically designed to retain only storms with measurable impact. This filtering reduces the number of storm events based not only on their physical characteristics (such as the maximum wind gust within their footprint) but also on the number of claims that can be linked to them.

The comments you raised underlines the complexity of t working with physical storms events. We will clarify this in the revised version of the manuscript.

Section 2.2

except if I missed it, Figure 1 is not referenced in the text. Please also provide dates in the caption or within the figure (if these are the tracks of named storms, that would be also nice to know). Also it would be useful to provide a reference in the map for 7.5 degrees West.

We will add the name of the storms are well as the 7.5W reference line (see Figure below). We will also add a reference to this figure when introducing the concept of cluster of storms.
Note that the cluster used for the illustration (Cluster 67) shows an example with 2 "storms" associated with few claims. In fact, they are respectively association with 120 and 70, however it's clear from the location of the claims that they are coming from 2 separate events. In fact, the claims associated to the depression of the 2006-12-30 are located in the North of France, which corresponds to the area were intense wind have been recorded.

[Figure]

*Figure 4 - Example of clustering with two storms. The green and blue lines represent the storm tracks of storms 1 and 2, with 6-hourly time increments. The green and blue shadings represent, respectively, the impact area of storms 1 and 2, defined with a radius of 1300 km around the centre of the track. The red shading illustrates the intersection of "high-impact" areas, different by a radius of 700 km around the storm tracks.*

[Figure]

*Figure 3- Association of claims to storms. Each column contains the maximum wind gust speed and the number of claims associated with a storm in Dec. 1999. Panels (a) and (c) show maps obtained for storm "2006-12-30 12h [-7.9;50.0]"and (b) and (d) for storm "2007-01-01 12h [-8.6;49.1]". Panels (a) and (b) show wind-gust footprints and the storm trajectories (thick black lines), panels (c) and(d) show the spatial distribution of the number of claims associated with each storm, while titles indicate the total number of claims for the whole event.*

Lines 127-128: Distance is 420 km (assuming that 70 km refers to 1 degree..). But is 96 hours a realistic timescale for a cluster of storms hitting the same place? Please elaborate. The 96h criterion is explained in section 5 but this comes too late.

We will explain the meaning of the 96-hour criterion in the introduction, highlighting that it is intended to align with existing reinsurance policies. The remarkable winter of 1999, marked by the successive storms Lothar and Martin, illustrates the realistic possibility of multiple intense storms striking the same region within a 96-hour period.

Lines 130-134. While I appreciate the numbers provided here, could we have a visual impression of the frequency of areas affected by storm clusters in France. Also, in connection with the above, could we have a figure with statistical information about the storms included in every cluster?

We will add in the annexes of the revised manuscript, the Figure 1 of this review-answer to illustrate the area most impacted by the clustered events. We can see in this figure that the storms being clustered and resulting in an impact for Generali are similar to the set of storms associated with impact.

2b. Attribution procedure

Section 3.1 was rather difficult to understand. Probably I get the big picture but the text is rather complex, dense and in some points rather incomprehensible (at least to me). I would certainly appreciate having an illustrative example (maybe also a flow diagram?) where each step of the procedure is shown in a bit more detail. Maybe the examples in section 4 could serve this cause.

We will add the following flow-diagram to clarify the different steps of the association procedure. The diagram was done for the case study of mid-January 2025 which contain the cluster of storm Klaus.

[Figure]

*Figure 5 - Flowchart of the association method for the case study of mid-January 2009.*

Lines 173-174 are quite cryptic to me. What does it mean "understanding of storm damage and Generali's exposure"? And what would it mean "few claims"? Please elaborate. How does this affect the results in this study. How representative is the exposure of Generali for the total of assets in France? Can your results be generalized about storm impacts in France? Actually, why the minimum number of claims is important here? Maybe one claim alone in an area where Generalli has a low exposure still insinuates an impactful event(?).

What we meant here is that storms are known to be large physical events leading to impact over large areas (as opposed to hail of flood hazards). When such events are detected, we can expect more than 10 houses impacted. This can be put in perspective with Generali's exposure. Generali France owes approximately 1million contracts over France, which represent 3% of the market share in France. The percentage of the market share can seem relatively small but in France, property

damage insurance, is mandatory for renters, co-ownerships and for owner if they need a loan. Hence we can assume that almost 100% of properties are insured. We will clarify this in the text.

Generali does not hold a homogeneous risk over France and is more exposed in some area, such as the South-Est or North of France. Nonetheless it's highly unlikely that a storm event will result in some damage in less that 50 properties.
Additionally, for reinsurance perspectives, storms event a reasonable number of claims. While this number is arbitrary, we decided to set it to 50 as it was minimizing the best the cost function defined in section 3.2.

I actually failed to understand several concepts in section 3.1. In a rather naive approach, I would say that since you already have the coordinates of the circular area influenced by every track point and since you already have the information on a claim, then you could simply say that if the claim spatially and temporally coincides with a storm-affected area (+/- a certain amount of time), then the claim is attributed to that track point. Of course this way, the claim could be attributed to more than one tracked features. In this case, the claim could be certainly attributed to a cluster of tracks (or storms). But I do not see the feasibility of the "..ultimate goal to associate each claim with a single storm" as stated in lines 171-172. Supposing that two track points are very close to each other and correspond to two different storms, or maybe they correspond to the tracks of a "bigger" storm with two centers. Then, the high wind gusts somewhere between two track points will be due to the interaction of these two storm centers (e.g. two distinct storm centers close to each other may result in higher pressure gradients and thus higher wind speeds). There is no really a reason to say that the high gusts are due to one or the other track point. Even in such a case, one could e.g. attribute the claim to the closest track point of either storm. Given that claims and track points are pinpointed and temporally well-defined, I am not very sure I follow the complexity of the approach here, the meaningfulness of the "dstorm" variable and of the cost function. As stated before, I would appreciate illustrative examples to better understand their necessity and use.

We first stress that claims are well pinpointed but not temporally well-defined. It's visible on the added flowchart as well as on Figure 7 and 8 of the manuscript. The figures of the flowchart show that although, some claims peaks can be identified, claims are recorded almost every day. This makes the association to the correct storm complicated, in particular when the dataset of tracks contains numerous members. The 2nd figure of the flow chart underlines that if no constraint is added on the number of claims a storm should be associated to, we end up with numerous storms members, in particular, some are associated with less than 10 claims. Given the number of policies held by Generali, this is extremely unlikely. Figure 7 also underlines the incorrect dating of claims. We can see claims declared on the 27, 28 (not shown be some claims were shown until the 30) over the area impacted by storm Martin.

Additionally, as you already mentioned earlier, the windstorm dataset contains numerous tracks, some of them being only weak low-pressure systems. As you also correctly pointed out, the radius of 1300km could be too big for such small events which would lead to contaminated footprints. In such cases, the footprints of small and weak depressions could include wind intensity from other storms. The final step of the association, which will group the claims around events with at least 50 claims will, in a sense, correct this contamination. In fact, final claims will be associated with the storm with the largest impact area. Tuning the $n_{claims}$ parameters will correct this potential contamination of storms footprint. A too small value would keep some potential overlapping of the footprint while a too large value will only identify the major events.

The implementation of the cost function ensures the robustness of the association. As explained in section 3.1, the association rely on 3 parameters. Two of them corresponds to the length of the day windows ($X_a$ and $X_b$), the last one is the intensity of the grouping over the major storm events ($n_{claims}$). This one can be tuned to aligned the best to the observed input claims, while keeping in mind that these are a bias representation of the actual damage caused by the storms. This was first done visually over specific winter with known successive cluster events (Anatol, Lothar, Martin in December 1999; Klaus in January 2009 but also Carmen and Eleanor in January 2018 which was not shown in the article). To asses more quantitatively and automatically the quality of the association, we designed 3 performances metrics. Again, we could tune them visually to see which combination of ($X_a, X_b, n_{claims}$) would be the best but might not lead to robust and reproducible result. The construction of the cost function ensures the finding of the best parameters without extensive visual arbitrary decisions. Also, we showed in Appendix A that the optimal parameters obtained are not too sensible to the constructed performance metrics. I agree that the whole process is technically heavy but it was needed to ensure the reproducibility and to justify the robustness of the association method.

We will clarify all of this in the revised manuscript.

Few additional comments

In the examples of Dec 1999, Line 350 states that misattribution compromises the vulnerability curves. I am not an expert but if I understand correctly the field, the vulnerability curve links wind speed with losses. So this is indifferent of a storm attribution procedure(?). If not, could you please explain a bit more this point?

Exactly: vulnerability curves associate potential loss ratio to wind speed intensities. They are usually built using observed damage but also building vulnerability features. A correct association of wind speed to loss ratios is crucial. Let's take a claim declared on the wrong day which was associated to a storm A but it was caused a storm B. The corresponding point for the calibration of the vulnerability curve would be to link the wind speed B to the loss ratios but if the claims are not attributed to the correct storm, the loss ratio would then be linked to the wind speed A, which might not be the correct one.
This could be illustrated with the case of Lothar and Martin. Let's take a claim declared around Bordeaux. If, based only on the date, the claim is associated to storm Lothar, it would be associated to a wind of approximately 30m/s. Alternatively, as a result of the association method, the claim is more likely to be associated to storm Martin and to wind values of about 38m/s. In the construction of the vulnerability curve, forecasting a given loss ratio for 30 or 38 m/s is drastically different.

The case above is of course idealized because we never know which of the two storms caused the damage and it could also be the combination of both which lead to such losses. The study of how the association procedure change the functions of the vulnerability curves, and how clustered events, with the succession of intense storms, should be treated in such curve, is the scope of another research.

In your methodological approach, if I understood correctly, "dstorm" is defined by coordinates which are located over the Ocean (7.5W). So choosing Xb = 3 days means that the claim could take place while a storm is really far from France (I guess still within 1300 kms of radius). Could you show examples where evidently high wind speed is relevant to a storm with a far-reached center? Maybe I misunderstood this part. Could you please clarify.

In fact, it might be counter intuitive to allow the window to take some days before the storm actually crossed the line of 7.5W but this is necessary due to the inherent bias in claim dates. As illustrated in the added flowchart (Figure 5 of this review-answer), claims are declared every day. When filling a claim request, the insured person can fill any date in the past. When the person does not know the exact date of the damage, which is usually the issue with such declaration, the person is as likely to fill a date before and after the actual damage. Hence this number of days before the storm crossed the line of 7.5W should not be viewed as an anticipation of a damage but more as a correction of the mistake potentially made during the declaration of the claim date. We will clarify this in the text.

As an additional note, please avoid using the verb "land" for stating that a track point is over continental areas (or a cyclone influences such). First of all, the coordinate of 7.5 W used here is over the sea, second, I presume that the use of this terminology is inspired by the field of tropical cyclones where the usual term is "landfall". But even so, "landfall" has a special weight because the highest wind speed is really close to the cyclone center. Therefore, time and area of landfalling cyclones is directly relevant to impacts, which is not -always- the case in ETCs.

We will rephrase this to "storm impact date". We will also clarify in the introduction that it's impossible to give a unique date to a storm event as it is a moving object. However, for comparison purposed, we decided to define the storm impact date as the date at which it crossed to longitude of 7.5W

Section 2.2 seems more like a "methods" rather than "Data" as stated in the title of section 2.

We will revise the name of this section

Section 3.1 Lines 161-167. This paragraph seems more adequate for the introduction.

We will consider adding this paragraph to the introduction

Copernicus C3S. (2025). *Windstorm tracks and footprints derived from reanalysis over Europe*

*between 1940 to present* [Jeu de données]. ECMWF. https://doi.org/10.24381/BF1F06A9

Dacre, H. F., & Pinto, J. G. (2020). Serial clustering of extratropical cyclones : A review of where, when

and why it occurs. *Npj Climate and Atmospheric Science*, *3*(1), 48.

https://doi.org/10.1038/s41612-020-00152-9

Flaounas, E., Aragão, L., Bernini, L., Dafis, S., Doiteau, B., Flocas, H., Gray, S. L., Karwat, A.,

Kouroutzoglou, J., Lionello, P., Miglietta, M. M., Pantillon, F., Pasquero, C., Patlakas, P.,

Picornell, M. Á., Porcù, F., Priestley, M. D. K., Reale, M., Roberts, M. J., … Ziv, B. (2023). A

composite approach to produce reference datasets for extratropical cyclone tracks :

Application to Mediterranean cyclones. *Weather and Climate Dynamics*, *4*(3), 639-661.

https://doi.org/10.5194/wcd-4-639-2023

Gramcianinov, C. B., Campos, R. M., De Camargo, R., Hodges, K. I., Guedes Soares, C., & Da Silva Dias, P. L. (2020). Analysis of Atlantic extratropical storm tracks characteristics in 41 years of ERA5 and CFSR/CFSv2 databases. *Ocean Engineering*, *216*, 108111. https://doi.org/10.1016/j.oceaneng.2020.108111

Hawcroft, M. K., Shaffrey, L. C., Hodges, K. I., & Dacre, H. F. (2012). How much Northern Hemisphere precipitation is associated with extratropical cyclones? *Geophysical Research Letters*, *39*(24), 2012GL053866. https://doi.org/10.1029/2012GL053866

Lockwood, J. F., Guentchev, G. S., Alabaster, A., Brown, S. J., Palin, E. J., Roberts, M. J., & Thornton, H. E. (2022). Using high-resolution global climate models from the PRIMAVERA project to create a European winter windstorm event set. *Natural Hazards and Earth System Sciences*, *22*(11), 3585-3606. https://doi.org/10.5194/nhess-22-3585-2022

Neu, U., Akperov, M. G., Bellenbaum, N., Benestad, R., Blender, R., Caballero, R., Cocozza, A., Dacre, H. F., Feng, Y., Fraedrich, K., Grieger, J., Gulev, S., Hanley, J., Hewson, T., Inatsu, M., Keay, K., Kew, S. F., Kindem, I., Leckebusch, G. C., … Wernli, H. (2013). IMILAST : A Community Effort to Intercompare Extratropical Cyclone Detection and Tracking Algorithms. *Bulletin of the American Meteorological Society*, *94*(4), 529-547. https://doi.org/10.1175/BAMS-D-11-00154.1

Parfitt, R., Czaja, A., & Seo, H. (2017). A simple diagnostic for the detection of atmospheric fronts. *Geophysical Research Letters*, *44*(9), 4351-4358. https://doi.org/10.1002/2017GL073662

Pinto, J. G., Gómara, I., Masato, G., Dacre, H. F., Woollings, T., & Caballero, R. (2014). Large-scale dynamics associated with clustering of extratropical cyclones affecting Western Europe. *Journal of Geophysical Research: Atmospheres*, *119*(24). https://doi.org/10.1002/2014JD022305

Priestley, M. D. K., Dacre, H. F., Shaffrey, L. C., Schemm, S., & Pinto, J. G. (2020). The role of secondary cyclones and cyclone families for the North Atlantic storm track and clustering

over western Europe. *Quarterly Journal of the Royal Meteorological Society*, *146*(728), 1184-1205. https://doi.org/10.1002/qj.3733

Priestley, M. D. K., Pinto, J. G., Dacre, H. F., & Shaffrey, L. C. (2017). Rossby wave breaking, the upper level jet, and serial clustering of extratropical cyclones in western Europe. *Geophysical Research Letters*, *44*(1), 514-521. https://doi.org/10.1002/2016GL071277

Renard, R. J., & Clarke, L. C. (1965). EXPERIMENTS IN NUMERICAL OBJECTIVE FRONTAL ANALYSIS[1]. *Monthly Weather Review*, *93*(9), 547-556. https://doi.org/10.1175/1520-0493(1965)093<0547:EINOFA>2.3.CO;2

Schemm, S., Rudeva, I., & Simmonds, I. (2015). Extratropical fronts in the lower troposphere–global perspectives obtained from two automated methods. *Quarterly Journal of the Royal Meteorological Society*, *141*(690), 1686-1698. https://doi.org/10.1002/qj.2471

Sinclair, V. A., & Catto, J. L. (2023). The relationship between extra-tropical cyclone intensity and precipitation in idealised current and future climates. *Weather and Climate Dynamics*, *4*(3), 567-589. https://doi.org/10.5194/wcd-4-567-2023

Vautard, R., Van Oldenborgh, G. J., Otto, F. E. L., Yiou, P., De Vries, H., Van Meijgaard, E., Stepek, A., Soubeyroux, J.-M., Philip, S., Kew, S. F., Costella, C., Singh, R., & Tebaldi, C. (2019). Human influence on European winter wind storms such as those of January 2018. *Earth System Dynamics*, *10*(2), 271-286. https://doi.org/10.5194/esd-10-271-2019

Zappa, G., Shaffrey, L. C., & Hodges, K. I. (2013). The Ability of CMIP5 Models to Simulate North Atlantic Extratropical Cyclones*. *Journal of Climate*, *26*(15), 5379-5396. https://doi.org/10.1175/JCLI-D-12-00501.1

---

## Author Comment (AC3)

This paper aims to (i) link insurance claims to individual ETCs, (ii) apply the method to clusters of ETCs and (iii) assess the impact of storm clusters on insured losses. Overall, the paper achieves the aims. In places the language used in the paper is overly emphatic for a piece of scientific writing and rewriting is needed to ensure that all statements are supported by evidence (see comments below for specific examples).

We thank the reviewer for the feedback. We appreciate the insightful and helpful comments which contribute to improve our manuscript. The responses to each of the comments appear in blue.

**General comments**

1. In many places the authors describe their work as fundamental, significant, valuable and a step forward. It should be left to the reader to evaluate the importance and novelty of the results presented. Please remove all emphatic adjectives.

We will remove these redundant emphatic adjectives.

2. There are several statements in the paper referring to small storms, however I could not find any description of how storm size was measured. The authors need to include this, or be more precise about what is meant by small storms, perhaps it refers to the intensity or duration of the storms rather than their physical dimensions?

We agree that "small storms" are frequently mentioned in the manuscript without being specifically defined. This initially refers to their intensity but also affect their physical extension. We will provide a characterization of these in the section 2.1.

3. The evidence to demonstrate the validity method is described in sections 4.2 and 4.3. This is after results using the method to link storm occurrence to losses (figure 6b, section 4.1). It would make more sense if the paper were reordered to demonstrate the method before it is used. Also, it would be helpful to provide quantitative information regarding the partitioning of losses between the storms in the cluster as well as partitioning the number of claims.

We take the recommandation and we will consider reorganizing the section 4. We will move the present section 4.1 to 4.3 and the case studies of sections 4.2 and 4.3 will be moved to sections 4.1 and 4.2. For these 2 case studies, we will add quantitative information about the partitioning of losses and number of claims, similar to information in figure 6.

**Specific comments**

4. Title: Is the word 'Unravelling' necessary in the title? This verbose language is common when using AI to suggest a title so should be avoided.

No AI chatbot was used in any part of this paper. The verb "Unravelling" corresponds exactly to what we want to highlight in the paper. Additionally, research on google scholar reveals that this verb has been used for climate science papers.

5. Line 5: The authors describe their own work as a cornerstone for insurance and reinsurance processes. This is a very bold statement and should be left to the reader to determine how fundamental the methodology presented is.

We will rephrase the sentence.

6. Line 19: Here and elsewhere the storms are described as being displaced, I think path or track might be a more familiar word to use.

We will rephrase the sentence.

Exactly: displacement refers to the track of the ETC. We will make this clearer.

7. Line 24: The SSI should be explained in more detail. What does it measure?

We will add a brief description of the SSI index.

8. Line 32: A Poisson distribution describes events that happen randomly and independently. I think that there is no reason why a Poisson distribution with a large mean cannot also have a large variance.

We will rephrase this sentence.

9. Line 44: The dispersion metric can be used globally I believe. Although I agree that it might not be suitable for impact assessments.

This is the point we want to raise. We will rephrase the sentence accordingly and remove the mention of "globally".

10. Line 55: How does the representation of hazards in insurance loss models lead to damage prevention?

We agree with the reviewer that the connection is not straightforward. A better representation of the hazard leads to a better comprehension of the risk. The insurer might then be more aware for the regions at risk and target the prevention accordingly in these specific areas. We will clarify this in the revised version of the manuscript.

11. Line 77: This sentence does not have an end. What advantages are unlocked? Perhaps 'unlocking several advantages' is not needed in this sentence?

The "several advantages" are listed in the following sentences for the different perspectives (vulnerability, meteorology). We will consider removing it or integrating it better with the rest of the paragraph.

12. Line 79: Where in the paper are the environmental factors leading to damage investigated?

We underlined, thanks to the association method, that clusters of storms are associated with exacerbated impact compared to individual storms. In particular, we state in line 280 "This means that storms can be concentrated over several days or weeks of a season and with given locations can experience both close successions or absences of storms." This represents an environmental factor exacerbating damage.
We acknowledge that the manuscript does not present any result on the structural factor contributing to damage. We will remove this part.

13. Line 79: Here the authors describe their work as 'significantly enhancing our understanding of storm clustering events'. This is a bold claim and perhaps best left to the reader to decide on the usefulness of the paper.

We will moderate this claim.

14. Line 83: The assessment of reliability is difficult since there is no truth with which to evaluate the method. Perhaps remove the word reliably?

Indeed, the manuscript aims to underline that there is no ground truth about which storm resulted in damage. We will remove the word "reliably".

15. Line 98: By displacement of trajectories, are the authors referring to the minimum storm track length?

We refer to the total length of the trajectory, both in terms of number of points and distance. We will modify the sentence to make this clearer.

16. Line 101: What is the impact distant from? The storm centre or metropolitan France?

The distance is always computed from the storm center. We will rephrase the sentences to make this more explicit.

17. Line 105: Closest point to what? Does this refer to the location along the storm track that occurs closest to a longitude of 7.5W?

Yes: the point of closest approach is measured by the distance between the storm tack points and the longitudinal line of 7.5W. We will clarify that sentence.

18. Line 110: It is not clear what the numbers in the curly brackets refer to.

The curly brackets refer to temporal window in number of hours around each track points. This means that for each track point, we will retain the maximal wind gust observed within the spatial mask of 1300km and temporal windows ranging from 12h before the date of the track point to 12h after.

19. Line 113: Earlier it is stated that a cyclone duration of 24hours is used, but here the authors state that the conditions on cyclone duration and intensity are 'relaxed'. What does this mean?

This means that fewer constraints are applied over the duration and intensity of the tracks, compared to the studies we are referring to in the paper. (Lockwood et al., 2022; Priestley et al., 2024) used the same TRACK algorithm but with a minimal cyclone duration of 48h (which is only 24h for us), a minimal displacement of the track of 1000km (which we did not constrain) and a minimal threshold of maximum vorticity of $1 * 10^{-5} s^{-1}$ (variable that we did not constrain either).

20. Lines 114 and 252: How do 'relaxed' constraints on cyclone duration and intensity refer to the speed and size of the storms considered? Additionally, on line 252 the authors refer to small storms, how is storm size determined?

Fast moving cyclones can cross the country and result in some damage in less than 48h, if they are detected in their late development. If conditions regarding the minimal duration are applied, such storms are likely to be discarded.

Regarding the usage of "small storms", we agree that this term is too vague for the objects we are trying to characterize. We will remove this in the revised version.

21. Line 118: I don't think the authors have included any evidence to support their statement that including storms 'of all scales' reduces the bias. What bias are they referring to and how have they determined the scale of the storms?

The bias is not used for it is statistical meaning. Here we refer to the fact that if the input set of tracks is too restrictive, the association method will try to align with it. In particular, over the 344 storms associated with some damage for Generali presented in the manuscript, 107 of the them would have been omitted with constrains identical to the ones selected by (Lockwood et al., 2022; Priestley et al., 2024). Pre-constraining the set of storm tracks is equivalent to say that only some type of storms can result in some claims for the insurance, which is not the case. The bias is the extent to which the impacting are conditioned by the set of storms tracks.

> 22. Line 124: Earlier the radius of the storms considered for impact was set to 1300km but here it is reduced to 700km. Which is used in the study and why are 2 radii for impact mentioned?

Several radii of influence can be found in the study of ETC. Paper working with frontal structure, for example associating precipitations to ETC, usually chose large radius of 12° (Hawcroft et al., 2012; Sinclair & Catto, 2023). On the other hand, some studies have used radius of 6° to characterize the area of strongest wind related to the ETC (Gramcianinov et al., 2020; Zappa et al., 2013).

The radius of 1300km used for the construction of the footprint was deliberately chosen to be large in order to capture the potential wind impact which, because of frontal structures, can be distant from the center of the ETC. Conversely, cluster of storms should be events for which common area could have been impact be successive storms. For this reason, after trial and error, we decided to the smaller radius of 700km. This also ensure that not all the storms are parts of clusters. We will clarify theses points in the revised version of the manuscript.

> 23. Line 177: How are the robustness and reliability of the results quantified?

Robustness and reliability are estimated using the 3 performances metrics presented in section 3.2. The robustness of the association is also evaluated with respect to theses metrics. Figure A1 shows that the optimal values of the tuning parameters $(X_a, X_b, n_{claims})$ is constant for a varying weight varying between 0.3 and 0.6.

> 24. Line 187: What does the closest local maximum refer to?

The closest local maximum corresponds the local maximum for which the date is the closest to the date of the storm. We will clarify this in the manuscript.

> 25. Line 188: What does the number of local maxima refer to?

This corresponds to the number of local maxima which are defined in line 181 as "The local maxima are identified by peaks over the time series of claim count gathering at least 10 claims."

> 26. Line 201: Here the authors state that lighter colours indicate better results. What is meant by better and is this the case for all 3 tuneable parameters?

Lighter color indicates better results for each of the 3 tunable parameters. "Better" has a different meaning for each parameter; this is described in section 3.2, in the sections between lines 202 and 221.

> 27. Line 237: How are the authors quantifying the accuracy/preciseness of their method? What are they comparing the method to?

Before the proposed association, no storm catalogue tailored to Generali's damage and exposure existed. The only available resources were global dataset of storm tracks or set of storms resulting in important impact, such as the ones presented in the introduction (line 65-68). The results are

compared global dataset which gather impact at country level and are not able to differentiate between successive events. The proposed method is thus more accurate because we manage to link damage to specific storms events and more precise because such association has been done at claim resolution. We will clarity this in the revised version of the manuscript.

28. Line 248: Here and in 12 other locations the authors use the phrase 'underlines'. This is quite repetitive use of language and alternative words could be used.

We understand that the referee finds this boring, but it is often recommended that scientific papers avoid the multiplication of synonyms (Day & Gastel, 2014).

29. Line 259: Figure 4 does not show the intensity of storms or the vulnerability, so it is not clear what the authors are using to support this statement.

Here, the intensity of the storm refers to the potential damage it has led to, which is visible in figure 4 with the total cost. We will rephrase this to make it clearer.

30. Line 263: Shift usually means a change. I believe the authors are simply referring to a difference here.

Yes, we will replace "shift" with "difference".

31. Line 266: What does 'cst' mean? Is this euro cents?

It refers to "constant euro", which is introduced in section 2.2, line 145. We will also introduce the "cst €" notation there for clarity.

32. Line 268: What is restricted about the winter 2018/2019 analysed by Mision Risques Naturels?

Here we meant that the analysis is restricted because it was only performed over one winter. We will rephrase this.

33. Line 276: How is the impact of the clusters measured? What criteria is used to class clusters as high-impact?

The impact is measured by the number of claims and/or the total losses associated with each storm of the cluster. A "high-impact" cluster is a cluster for which at least 2 of the storms have been associated with damage for Generali. This will be clarified in the revised text.

34. Line 277: Is an impacting storm one that resulted in loses for Generali only or one which has a track within 1300km of France?

An impacting storm is a storm which have been associated with claims for Generali. We will emphasize that point.

35. Line 302: What is meant by 'far from the 100%'?

It means that total losses of the cluster cannot be assumed to be hold by only the costliest storm as this one represents only 72% (on average) of the total losses associated to the cluster. We will rephrase that more clearly.

36. Lines 312-324: These paragraphs repeat what is describes in earlier paragraphs so should be removed.

Indeed, this is a duplicate. We will remove it.

37. Lines 397-403: This paragraph is speculative. It does not describe the results from the paper so should be reworded or removed.

In fact, this was not directly addressed in the paper. We will move it to section 5.

38. Line 417: What is meant by a gathering around storms with at least 50 claims?

It corresponds to the last step of the method described in section 3.1, done with the $n_{claims}$ parameter.

39. Line 419: What is meant by 'highly transposable to other outlooks'? I am unsure what an outlook is.

We meant that this method can be used for other hazards (flood, convective storms…) but also using other types of damage (hail, water infiltration).

40. Line 443: By 'came alone' are the authors referring to non-clustered storm events?

Yes, we are referring to the storms which were not part of clusters. We will rewrite this.

41. Line 474: The authors evaluate their framework as valuable. This evaluation should be left to the reader.

We will remove this subjective statement.

42. Line 484: The authors describe their findings are a step forward in understanding. This evaluation should be left to the reader.

We will remove this subjective statement.

**Typographical errors**

1. Line 23: The sentences describing windstorm metrics should form a separate paragraph since it introduces a new idea to the sentences preceding them.

We will write separate paragraphs.

2. Line 47: Do you need both event and impact here?

We will only keep "impact-perspective"

3. Line 65: Here 'including' should be 'such as'.

We will modify this.

4. Line 89: I think 'over' should be 'including'?

We will modify this.

5. Line 93: ERA stands for ECMWF ReAnalysis so it is not necessary to include the words 'historical reanalysis' afterwards.

We will rewrite theses occurrences.

6. Line 103: I think 'landfall' is more commonly used than 'landing'.

Actually "landfall" mainly refers to tropical cyclones. We will change "landing date" to "impact date".

7. Line 127: Should 70km be 700km to be consistent with line 124?

Yes,  thanks you.

8. Line 149: The 'st' in italics should be roman text.

Indeed, we will fix this.

9. Line 203: What is the 'peal date'?

I should be "local maxima".

10. Line 354: Why is approx. in italics?

This will be rewritten to roman text.

11. Line 426: Why is the word 'soley' included in this sentence?

We will remove this word.

12. Line 449: Why is an 'double hyphen used here?

We will remove the double hyphen which is not needed.

Cornér, J. S., Bouvier, C. G. F., Doiteau, B., Pantillon, F., & Sinclair, V. A. (2024). *Classification of North Atlantic and European extratropical cyclones using multiple measures of intensity*. Atmospheric, Meteorological and Climatological Hazards. https://doi.org/10.5194/egusphere-2024-1749

Day, R. A., & Gastel, B. (2014). *How to write and publish a scientific paper* (7. ed., 4. print). Cambridge Univ. Press.

Gramcianinov, C. B., Campos, R. M., De Camargo, R., Hodges, K. I., Guedes Soares, C., & Da Silva Dias, P. L. (2020). Analysis of Atlantic extratropical storm tracks characteristics in 41 years of ERA5 and CFSR/CFSv2 databases. *Ocean Engineering*, *216*, 108111. https://doi.org/10.1016/j.oceaneng.2020.108111

Hawcroft, M. K., Shaffrey, L. C., Hodges, K. I., & Dacre, H. F. (2012). How much Northern Hemisphere precipitation is associated with extratropical cyclones? *Geophysical Research Letters*, *39*(24), 2012GL053866. https://doi.org/10.1029/2012GL053866

Lockwood, J. F., Guentchev, G. S., Alabaster, A., Brown, S. J., Palin, E. J., Roberts, M. J., & Thornton, H. E. (2022). Using high-resolution global climate models from the PRIMAVERA project to create a European winter windstorm event set. *Natural Hazards and Earth System Sciences*, *22*(11), 3585-3606. https://doi.org/10.5194/nhess-22-3585-2022

Priestley, M. D. K., Stephenson, D. B., Scaife, A. A., Bannister, D., Allen, C. J. T., & Wilkie, D. (2024). Forced

trends and internal variability in climate change projections of extreme European windstorm

frequency and severity. *Quarterly Journal of the Royal Meteorological Society*, *150*(765), 4933-4950.

https://doi.org/10.1002/qj.4849

Sinclair, V. A., & Catto, J. L. (2023). The relationship between extra-tropical cyclone intensity and precipitation

in idealised current and future climates. *Weather and Climate Dynamics*, *4*(3), 567-589.

https://doi.org/10.5194/wcd-4-567-2023

Zappa, G., Shaffrey, L. C., & Hodges, K. I. (2013). The Ability of CMIP5 Models to Simulate North Atlantic

Extratropical Cyclones*. *Journal of Climate*, *26*(15), 5379-5396. https://doi.org/10.1175/JCLI-D-12-

00501.1